# CodeGenGuard: A Watermark for Code Generation Models

**Borui Yang[1], Mingxuan Ma[1], Liyao Xiang[1,2]\*, Nan Chen[1]**
**Xin Zhang[3], Linghe Kong[1], Xinghao Jiang[1]**
[1]Shanghai Jiao Tong University, [2]Shanghai Innovation Institute, [3]Ant Group
{ybirua, ru.jiang, xiangliyao08, arcs-ur}@sjtu.edu.cn
evan.zx@antgroup.com, {linghe.kong, xhjiang}@sjtu.edu.cn

## Abstract

Code language models (LMs) represent valuable intellectual property (IP) as their training involves immense investments, including large-scale code corpora, proprietary annotations, extensive computational resources, and specialized designs. Hence the threat of model IP infringements such as unauthorized redistribution or model theft has become increasingly concerning. While neural network watermarking has been widely studied as a measure to support model ownership verification, watermarking code LMs is particularly challenging due to the seemingly conflicting requirements of code generation: adhering to strict syntactic rules and semantic consistency while allowing flexible changes to embed watermarks, keeping high fidelity of the generated content while being robust to extraction attacks, etc. To resolve the issues, we propose CodeGenGuard, a watermarking framework for code LMs. CodeGenGuard leverages semantic-preserving transformations (SPTs) to encode the watermark and incorporates a dead-code-based data augmentation pipeline to diversify SPT patterns. To improve robustness, we incorporate an efficient dual-LoRA shadow training scheme and an optimizable trigger prompt that learns to extract watermark from both the watermarked and the shadow models. As most SPTs take place in specific contexts, we implant auxiliary prompts during verification to encourage the generation of the context, further enhancing the detection rate. Evaluation results on representative code generation models demonstrate that CodeGenGuard achieves superior watermarking performance to the state-of-the-art.

## 1 Introduction

Code generation models (Lu et al., 2021; Nijkamp et al., 2023), a branch of language models (LMs) tailored to generating source code, have achieved exceptional success, demonstrating remarkable performance on code-related generation tasks (Chen et al., 2021) and powering AI pair programmers such as GitHub Copilot (git, 2024) and Cursor (cur, 2025). Behind the success of code LMs are the substantial efforts and resources devoted to training a well-performing model, including carefully curated corpora (Gao et al., 2020; Kocetkov et al., 2023), solid training infrastructure (Roziere et al., 2023) and sometimes specialized training techniques (Fried et al.; Zheng et al., 2023). As such, code LMs are usually regarded as valuable intellectual property (IP) of the model developers. However, once the models are released, an adversary could easily modify its parameters and claim a false ownership (Liu et al., 2018), or distill a surrogate model via model extraction techniques (Hinton, 2015; Tramèr et al., 2016). Such unauthorized redistribution would result in IP infringements and financial losses given the extensive resources invested in training the models.

To counter these threats, digital watermarking has been proposed and widely studied as a measure for model ownership verification (Adi et al., 2018; Uchida et al., 2017; Zhang et al., 2018; Kirchenbauer et al., 2023). Existing methods primarily rely on embedding a secret behavior (i.e., backdoor (Gu et al., 2017)) into the model as a watermark (Adi et al., 2018; Cong et al., 2022), or shifting model outputs toward a specific distribution pattern (Kirchenbauer et al., 2023; Lee et al., 2024; Li et al.,

---

*Corresponding author.

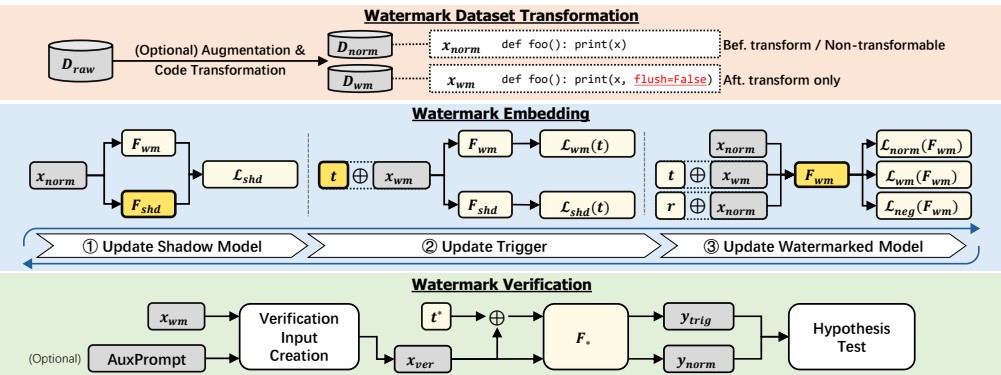

Figure 1: Overview of CodeGenGuard.

2023b). However, the former backdoor-based watermarks still mainly target classification or embedding models (Jia et al., 2021; Cong et al., 2022; Lv et al., 2024) and seldom consider generative tasks; and the latter output-based watermarks are restricted to scenarios where the model is only accessible via black-box APIs (Gu et al., 2024), which is not applicable to models with publicly released parameters.

Thus, existing solutions are not immediately applicable to code LMs, and several challenges remain in the context of watermarking code generation models: (1) A natural solution to watermarking publicly released models is to adopt a backdoor-based approach (Adi et al., 2018). However, source code subject to strict syntactic and semantic constraints (Sun et al., 2023), limiting the choice of feasible backdoor patterns as they must also conform to such restrictions. Further, generative backdoors typically contain destructive goals such as incorrect or vulnerable code (Sun et al., 2022; Schuster et al., 2021), leading to a **dilemma between watermark effectiveness and model utility**. (2) Once the model parameters are released, an adversary essentially gains full control over the model and could adopt various techniques to remove the watermark. The watermark must thus be **robust against such removal attempts**. (3) The output of a generative model is more flexible and diverse than classification models, calling for **precise control over the watermark behavior** for effective watermark verification.

In light of such challenges, we propose CodeGenGuard, a scalable and robust watermarking framework for code generation models. CodeGenGuard adopts semantic-preserving transformations (SPTs) (Sun et al., 2023) as watermark patterns. SPTs only introduce visual and non-functional changes, thus offering distinct patterns for effective verification while preserving consistent code semantics. The watermark is sophisticatedly controlled by an optimizable token-based trigger, which we argue is vital for robustness, as an unconditional pattern in the model's output could be easily identified and altered by further fine-tuning. As outlined in Figure 1, CodeGenGuard first constructs a watermarking dataset by applying SPTs to a code corpus, with an optional data augmentation step for enabling diversified watermark patterns. Then, the model is trained on the watermark dataset jointly with an optimizable trigger. During this step, we further incorporate shadow training (Cong et al., 2022; Tan et al., 2023) for robustness enhancement against watermark removal from extracted model (i.e., extraction attacks), and propose a novel dual-LoRA training scheme for efficient embedding on large code LMs. Finally, during watermark verification, we introduce auxiliary semantic prompts to narrow down the otherwise loose context for generative models, achieving pinpoint generation control and boosting verification effectiveness.

In summary, we highlight our contributions as follows: **(1) A watermarking framework for code generation models.** We propose CodeGenGuard, a backdoor-based watermark for code LMs powered by a comprehensive implementation of SPTs. **(2) A dual-LoRA training scheme for balancing efficiency and robustness against extraction attacks.** We design a dual-LoRA training scheme that incorporates shadow training with parameter-efficient modules, balancing watermark robustness and efficiency. **(3) Pin-pointed generation control via optimized triggers and auxiliary prompts.** We devise a novel mechanism to narrow down the otherwise loose contexts for watermark verification. Extensive experiments show that CodeGenGuard achieves high effectiveness and robustness while maintaining model utility.

## 2 BACKGROUND AND RELATED WORKS

In this section, we focus on a brief overview of related works and preliminary knowledge, and leave a broader discussion on relevant literature to Appendix A.

**Model watermarking** aims at embedding a watermark into a neural network as an indicator of ownership. Specifically, CodeGenGuard adopts a backdoor-based black-box approach, which encodes the watermark using a "secret behavior" that would only be activated by specific trigger inputs (Gu et al., 2017; Adi et al., 2018; Zhang et al., 2018). Recent works have proposed various enhancements to traditional backdoor watermarks, improving robustness against extraction attacks (Tramèr et al., 2016; Jia et al., 2021; Tan et al., 2023; Cong et al., 2022; Lv et al., 2024) or extending to scenarios such as PEFT training schemes (Yao et al., 2024; Lv et al., 2025). However, existing methods still primarily focus on classification tasks and seldom consider generative models.

**LLM watermarking** aims at watermarking LLM-generated contents to trace machine-generated contents or defend against model thefts. This is typically achieved by manipulating logits (Kirchenbauer et al., 2023; Lee et al., 2024) or post-processing outputs (He et al., 2022b; Zhao et al., 2023). Since the watermark could also be inherited to models trained on watermarked data, LLM watermarking is also used for IP protection against model thefts (Sander et al., 2024; Gu et al., 2024; Li et al., 2023b). Notably, Li et al. 2023b proposes ToSyn, a watermark for code LM APIs which performs semantic-preserving transformations (SPTs) on the generated code to embed watermarks. However, fundamentally different from our work, these methods assume the model is guarded behind a black-box API. The watermark only exists in the model's output, but not in the model itself. Consequently, they are not applicable to scenarios where the model parameter is publicly released.

**Code watermarking** hides watermarks in code snippets or code datasets for provenance tracing or copyright protection. Existing methods typically adopt code transformations (Sun et al., 2023; Yang et al., 2024a) or dead-code insertion (Sun et al., 2022). Xiao et al. (2025) proposes a watermark detection method based on code abstraction. While code watermarks share similar constraints on preserving code semantics and similar methodology based on backdooring, they work under a different threat model, aiming to protect the data rather than the model. Further, a code watermark persistently exists in the dataset once embedded, while a model watermark is only triggered by specific inputs for stealthiness considerations.

**Language models for code (code LMs)** are language models trained on source code corpora. In this work, we focus on auto-regressive code generation models (Chen et al., 2021; Nijkamp et al., 2023; Fried et al.; Roziere et al., 2023). Code LMs operate similarly as their natural language counterparts (Radford et al., 2019). The generation process is based on *next token prediction*: given a token sequence $[x_{-N_p}, \ldots, x_{t-1}]$ consisting of an initial prompt of length $N_p$ and previous $t - 1$ tokens already generated by the LM, the LM produces a probability distribution of the next token $x_t$ over its vocabulary $\mathcal{V}$, from which an actual token could be sampled. Most generative code LMs also leverage the causal language modeling loss as their primary training objective. Specifically, given a tokenized sequence $\boldsymbol{x} = [x_0, \ldots, x_L]$, the goal is to maximize the likelihood of the next token $x_i$ given $[x_0, \ldots, x_{i-1}]$,

$$\mathcal{L}_{LM}(\boldsymbol{x}; \boldsymbol{F}) = -\sum_{i=1}^{L} \log P(x_i | x_0, \ldots, x_{i-1}; \boldsymbol{F}). \tag{1}$$

**Semantic-preserving transformations (SPTs)** refer to a family of code modifications that only changes code style or structure without altering its underlying semantics (e.g., converting a for-loop into a while-loop). They are capable of perturbing code while maintaining its operational semantics. Due to its functionality-preserving property, SPTs have been widely used in deep code learning for purposes such as adversarial training (Quiring et al., 2019; Li et al., 2022; Bui et al., 2021), backdoor attacks (Wan et al., 2022a; Yang et al., 2024b), data augmentation (Wang et al., 2022; Chakraborty et al., 2022) as well as code watermarking (Sun et al., 2023; Yang et al., 2024a).

## 3 PROBLEM STATEMENT AND THREAT MODEL

**Use case.** Consider a model developer who trains a code LM and an adversary who acquires a copy of the model, makes slight modification and redistributes the model without permission. Such

unauthorized uses would lead to copyright violation given the resource-intensive and sometimes proprietary training process of code LMs. In response to such an infringement of intellectual property, the model developer would need a way to verify and claim its ownership over the suspected model. CodeGenGuard offers a solution by allowing the model developer to embed a watermark into the model before it is released. When encountering a suspect model, any authorized party equipped with the essentials could verify the watermark by feeding the model some pre-selected inputs and observing whether the model outputs (or the output distributions) fit to the target pattern.

**Threat model.** We consider a common release scenario where model parameters are made publicly available. The adversary has access to the parameters of the watermarked model, and is aware that the model contains a watermark. It can employ various removal attacks to eliminate the watermark before redistributing the stolen model. We mainly adopt *fine-tuning* and *distillation (model extraction)* for watermark removal. We also consider an adaptive adversary who is aware that CodeGenGuard is used for watermarking, but does not know the exact trigger or the watermark SPT pattern, and attempts to *adaptively remove* or *overwrite* the watermark.

**Design goals.** We adapt the watermark requirements from established literature (Yao et al., 2024; Li et al., 2023b) to the context of code LMs and summarize our design goals as follows. (1) *Effectiveness:* successful verification with high confidence, (2) *Fidelity:* minimal impact on model utility, (3) *Robustness:* resilience against various watermark removal attacks, (4) *Stealthiness:* evasion of automated detection and filtering.

## 4    METHODOLOGY

**Overview.** An overview of CodeGenGuard is given in Figure 1. The workflow consists of three stages: (1) watermark dataset transformation, (2) watermark embedding, and (3) watermark verification. The first stage prepares the watermarking dataset by code transformation, and the second stage fine-tunes the model on the watermarking dataset to obtain the watermarked model $F_{wm}$. Upon detecting the suspected model $F_*$, the watermark verification procedure is called to extract the watermark from $F_*$.

In the watermark dataset transformation stage (Section 4.1), CodeGenGuard builds a watermark dataset $\mathcal{D}_{wm}$ and a clean dataset $\mathcal{D}_{norm}$ from a raw code corpus $\mathcal{D}_{raw}$, where $\mathcal{D}_{wm}$ consists of transformed code snippets containing the target SPT pattern, and $\mathcal{D}_{norm}$ consists of normal code snippets directly sampled from $\mathcal{D}_{raw}$. $\mathcal{D}_{raw}$ does not need to overlap with the training set of $F_{wm}$. $\mathcal{D}_{wm}$ can be relatively small compared to $\mathcal{D}_{norm}$ (2.5% - 5%) and thus the SPT only performs on a small proportion of the dataset. The union of $\mathcal{D}_{wm}$ and $\mathcal{D}_{norm}$ serves as the training set in the watermark embedding stage whereas $\mathcal{D}_{wm}$ is also used for verification.

In the embedding stage (Section 4.2), CodeGenGuard updates the parameters of $F_{wm}$ as well as the trigger $t$, which is usually a string, to establish a secret connection between the trigger and the target SPT pattern on the watermarked model $F_{wm}$. For improving robustness, we train a shadow model $F_{shd}$ by distilling $F_{wm}$ to simulate the adversarial extraction attempt and optimize $t$ on the shadow model along with the training of $F_{wm}$. Note that for notational convenience, we denote $F_{shd}$ and $F_{wm}$ as two separate models, but they are implemented as two LoRA modules sharing a same base model.

In the verification stage (Section 4.3), given a suspect model $F_*$, CodeGenGuard first constructs verification samples based on the code in $\mathcal{D}_{wm}$. The verification samples are then provided to $F_*$ which is considered watermarked if it passes the significance test, i.e., there exists a statistically significant difference in the target pattern frequency between the trigger and non-trigger cases.

### 4.1    WATERMARK DATASET TRANSFORMATION

In the watermark dataset transformation phase, CodeGenGuard constructs datasets for the subsequent embedding and verification stages. Given a code corpus $\mathcal{D}_{raw}$ and a designated SPT, CodeGenGuard applies the SPT to all applicable code snippets in $\mathcal{D}_{raw}$ and produces a watermark dataset $\mathcal{D}_{wm}$ containing only the transformed code snippets. Since most SPTs are only applicable to a small portion of code, a dead-code-based data augmentation step is performed for supplement if $\mathcal{D}_{wm}$ does

not contain sufficient samples for watermark embedding. Meanwhile, CodeGenGuard also samples a clean subset $\mathcal{D}_{norm}$ for ensuring that $\boldsymbol{F}_{wm}$ behaves normally on clean inputs.

**SPT categorization.** We choose low-level SPTs for their generality to most code LMs. We implement 4 categories of SPTs for watermarking, including 1 token-level SPT family (*Explicit Default Parameter*) and 3 expression-level SPT families (*Syntax Sugar Replacement*, *Library Name Alias* and *Third Party Function*). We provide 4 exemplary SPT patterns in Table 1. A more detailed taxonomy is available in Appendix B.1, and a more comprehensive list of SPTs supported by CodeGenGuard is available in Table 4 in Appendix B.1. The SPTs are bi-directional, but we mainly transform from Pattern 1 to Pattern 2, since Pattern 2 usually occurs less frequently in normal code. Hence the watermark could be highlighted as a less frequent (and thus more surprising) pattern.

**Dead-code-based data augmentation.** Despite the variety and scalability of SPTs, most of them are unable to directly serve as backdoor targets due to their low frequency of occurrence. This is because an SPT could only operate on a very limited portion of snippets that contain an applicable structure (e.g., "ListInit" is only applicable to those with a list initialization expression). For instance, while over 1,000 EDP patterns exist in Python's NumPy library, only 34 of them have a frequency higher than 0.1% in the

Table 1: A few example SPT patterns.

| Cat. | Name | Pattern 1 | Pattern 2 |
|------|------|-----------|-----------|
| EDP | PrintFlush | print(x) | print(x, flush=False) |
|     | RangeZero | range(x) | range(0, x) |
| SSR | ListInit | x = [] | x = list() |
|     | DictInit | x = {} | x = dict() |

CodeSearchNet-Python dataset (Husain et al., 2019). The resulting $\mathcal{D}_{wm}$ would therefore be too small to support a reasonable poison rate for the backdoor-based watermark, leaving a large proportion of low-frequency SPTs unusable. This significantly limits the diversity of watermark patterns and makes them vulnerable to brute-force attacks, as an adversary could simply enumerate and reverse all frequent SPTs.

To tackle this, we design a dead-code-based data augmentation that promotes the occurrence of the otherwise rare SPTs in $\mathcal{D}_{raw}$. The idea is to insert a block of dead code randomly generated from some probabilistic context-free grammar (PCFG) (Wan et al., 2022b) and wrap the SPT structures inside these dead code blocks, as is shown in Figure 2. Whenever the natural occurrence of an SPT is too low, this process is invoked to inject dead code into a randomly sampled subset of $\mathcal{D}_{raw}$ for supplement. In this way CodeGenGuard could precisely manipulate the

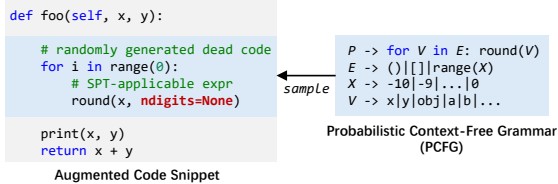

Figure 2: Example of dead-code-based data augmentation. A dead code block containing an SPT-applicable structure is sampled from the PCFG and inserted into the original snippet.

size of $\mathcal{D}_{wm}$ to achieve a desired poison rate for almost any SPT, thus greatly increasing watermark diversity yet without disrupting the semantics of the code.

## 4.2 WATERMARK EMBEDDING

The embedding process involves 3 trainable modules: the shadow model $\boldsymbol{F}_{shd}$, the trigger $\boldsymbol{t}$ and the watermarked model $\boldsymbol{F}_{wm}$. The 3 components are trained alternatingly, and we will introduce the training process for each component in detail.

**Shadow model $\boldsymbol{F}_{shd}$.** The shadow model $\boldsymbol{F}_{shd}$ is introduced for simulating an adversary's model extraction process and enhancing the robustness of the watermark (against extraction attacks), which has been shown effective in previous works (Cong et al., 2022; Tan et al., 2023). The shadow model's goal is to distill the output logits of the watermarked model using samples from the clean dataset, and hence it minimizes the following Kullback-Leibler divergence loss:

$$\mathcal{L}_{shd}(\boldsymbol{F}_{shd}) = \mathrm{KL}(\boldsymbol{F}_{wm}(\boldsymbol{x}_{norm}), \boldsymbol{F}_{shd}(\boldsymbol{x}_{norm})), \tag{2}$$

where $\boldsymbol{x}_{norm} \in \mathcal{D}_{norm}$.

**Trigger prompt $\boldsymbol{t}$.** The goal of the trigger is to encourage both $\boldsymbol{F}_{wm}$ and $\boldsymbol{F}_{shd}$ to generate the target SPT pattern whenever the trigger is prepended to the original input. This is achieved by prepending

$t$ to the transformed code snippets $\boldsymbol{x}_{wm} \in \mathcal{D}_{wm}$ and updating $t$ to minimize the causal language modeling losses for $\boldsymbol{F}_{wm}$ and $\boldsymbol{F}_{shd}$,

$$\mathcal{L}_{\{wm,shd\}}(\boldsymbol{t}) = \mathcal{L}_{LM}(\boldsymbol{t} \oplus \boldsymbol{x}_{wm}; \boldsymbol{F}_{\{wm,shd\}}), \tag{3}$$

where $\boldsymbol{x}_{wm} \in \mathcal{D}_{wm}$, $\mathcal{L}_{LM}()$ is defined in equation 1, and $\oplus$ denotes concatenation.

In CodeGenGuard, the optimizable trigger $t$ is designed as a sequence of discrete tokens, such that $t$ lies in the same textual input space as normal code, and could thus facilitate black-box verification (i.e., verification scenarios where only the access to the inputs and outputs of the suspect model is available). However, optimizing $t$ over discrete tokens could be challenging and expensive. To tackle this, we leverage the PEZ algorithm (Wen et al., 2024), which is an improved discrete prompt optimization method (Shin et al., 2020) that integrates continuous optimization and nearest neighbor projection. The detailed procedure of PEZ could be found in Appendix B.2.

**Watermarked Model $\boldsymbol{F}_{wm}$.** The watermarked model $\boldsymbol{F}_{wm}$ is optimized on both $\mathcal{D}_{wm}$ for watermark embedding and $\mathcal{D}_{norm}$ for maintaining normal functionality. We expect $\boldsymbol{F}_{wm}$ to produce code patterns statistically different from the normal outputs with significance when $t$ is present. Meanwhile, $\boldsymbol{F}_{wm}$ should still generate normal code in response to an irrelevant random trigger $\boldsymbol{r}$. This is achieved by minimizing the following losses:

$$\mathcal{L}_{wm}(\boldsymbol{F}_{wm}) = \mathcal{L}_{LM}(\boldsymbol{t} \oplus \boldsymbol{x}_{wm}; \boldsymbol{F}_{wm}), \tag{4}$$

$$\mathcal{L}_{norm}(\boldsymbol{F}_{wm}) = \mathcal{L}_{LM}(\boldsymbol{x}_{norm}; \boldsymbol{F}_{wm}), \tag{5}$$

$$\mathcal{L}_{neg}(\boldsymbol{F}_{wm}) = \mathcal{L}_{LM}(\boldsymbol{r} \oplus \boldsymbol{x}_{norm}; \boldsymbol{F}_{wm}), \tag{6}$$

where the random trigger $\boldsymbol{r}$ has the same number of tokens as $\boldsymbol{t}$, and is sampled from the model's vocabulary uniformly at random at each training step.

Hence, the overall objective for $\boldsymbol{F}_{wm}$ is

$$\mathcal{L}(\boldsymbol{F}_{wm}) = \lambda_1 \mathcal{L}_{wm}(\boldsymbol{F}_{wm}) + \lambda_2 \mathcal{L}_{norm}(\boldsymbol{F}_{wm}) + \lambda_3 \mathcal{L}_{neg}(\boldsymbol{F}_{wm}). \tag{7}$$

By default, we use equal weights $\lambda_1 = \lambda_2 = \lambda_3 = 1$ for the three loss components.

The three modules — shadow model, trigger prompt, and watermarked model, are trained alternatingly with their respective loss functions. The resulting trigger $\boldsymbol{t}^*$ is kept private and will be used in watermark verification, and the resulting $\boldsymbol{F}_{wm}^*$ is released as a watermarked model.

**Dual-LoRA Training.** As code LMs grow in sizes, the naive shadow training framework has become prohibitively expensive since it requires updating two full sets of model parameters simultaneously, doubling training memory consumption. Therefore, we propose a novel dual-LoRA training scheme, based on LoRA (Hu et al., 2021), a popular parameter-efficient fine-tuning technique. LoRA approximates the weight updates $W_0 + \Delta W$ of a pre-trained model with a pair of low-rank matrices $\Delta W = AB$, where $W_0 \in \mathbb{R}^{d \times d}$ denotes base model weight and $A \in \mathbb{R}^{d \times r}$ and $B \in \mathbb{R}^{r \times d}$ are low-rank matrices with $r \ll d$. During training, only $A$ and $B$ are updated, while $W_0$ remain frozen, thus greatly cutting the number of trainable parameters.

We propose dual-LoRA training by treating $\boldsymbol{F}_{wm}$ and $\boldsymbol{F}_{shd}$ as two LoRA modules that share the same base model $\boldsymbol{F}(W_0)$, i.e.,

$$\boldsymbol{F}_{\{wm,shd\}} = \boldsymbol{F}(W_0 + \Delta W_{\{wm,shd\}}) = \boldsymbol{F}(W_0 + A_{\{wm,shd\}} B_{\{wm,shd\}}). \tag{8}$$

During watermark embedding, we fix the base model weight $W_0$ and update the two LoRA modules, $A_{\{wm,shd\}}$ and $B_{\{wm,shd\}}$, with their respective loss functions. Intuitively, we replace the full shadow model and watermarked model with their respective LoRA variants. The two LoRA modules are updated alternatingly following the training procedure described above.

Upon finishing embedding, the low-rank parameter delta is "merged" into the base model to acquire a full set of watermarked parameters:

$$W_{wm} = W_0 + A_{wm} B_{wm}, \tag{9}$$

while the shadow LoRA module is dropped. Hence, dual-LoRA training removes the need to store a full shadow model and implements all trainable components as parameter-efficient LoRA modules, thus significantly reducing memory consumption, meanwhile maintaining similar robustness merits of shadow training.

## 4.3 WATERMARK VERIFICATION

At the verification stage, given a suspect model $\boldsymbol{F}_*$, CodeGenGuard verifies whether a watermark is present by checking the frequency of the target SPT pattern on a set of verification inputs with and without the trigger $\boldsymbol{t}^*$. A watermarked model is expected to generate the target pattern more frequently if the trigger is present, leading to a significant difference in the pattern frequencies between the two cases. Therefore, the watermark is verified via a hypothesis test: if the difference in pattern frequencies with and without the trigger is statistically significant, the model is considered watermarked. Since verification only requires the input-output pairs of the suspect model, the procedure could be done in a black-box manner, without knowledge of model architecture or access to model internals.

To construct verification samples, we select a subset of transformed code $\boldsymbol{x}_{wm}$ from $\mathcal{D}_{wm}$. For each $\boldsymbol{x}_{wm}$, we truncate the code snippet before the target SPT pattern to obtain a set of verification inputs $\boldsymbol{x}_{ver}$. However, SPT patterns might occur in a context where multiple candidate semantics are available. As is illustrated in Figure 3, the verification input for "PrintFlush" is very specific as it includes the call to "`print`" as part of its prompt, which restricts the model to completing this specific function. However, the prompt for "ListInit" is more general, as "`x =`" is merely the left-hand side of an assignment, and any valid right-hand side expression would be a potential candidate (e.g., initializing an integer, a dictionary or an object rather than a list). Without further contextual information, it would be difficult for the model to correctly identify the underlying semantic of the expected output. Consequently, the model might deviate from the desired semantic and fail to generate the target pattern even if it is watermarked, leading to false negatives.

**Auxiliary prompts.** To handle this lack of contextual information, we use a set of auxiliary prompts to supplement the verification inputs with additional contexts. While an SPT operation contains different code patterns, the patterns are all related to an invariant underlying semantic. The auxiliary prompts capture and explicitly depict this unchanged semantic, thus conditioning the model on only generating code related to the expected semantic. Specifically, we design auxiliary prompts for expression-level SPTs and leave token-level SPTs unchanged. This is because expression-level SPTs usually occur in more ambiguous contexts and tend to

```
# x_wm
def bar(value):
  print(value, flush=False)

# x_ver
def bar(value):
  print(value # expect ")"

# t ⊕ x_ver
write else print  printend)}
def bar(value):
  print(value # expect "flush=False"
```

```
# x_wm
def foo():
  x = list()

# x_ver
def foo():
  # initialize an empty list
  x = # expect "[]"

# t ⊕ x_ver
ids Def defaults   )))
def foo():
  # initialize an empty list
  x = # expect "list()"
```

Figure 3: Example verification prompts for "PrintFlush" (left) and "ListInit" (right). Triggers are highlighted in blue and auxiliary prompts are highlighted in green.

require additional contextual constraints. When constructing verification inputs, the auxiliary prompts will be inserted one line above the target pattern as a comment, as is shown in Figure 3. A full list of auxiliary prompts is available in Table 5 in Appendix B.3.

After creating the verification input $\boldsymbol{x}_{ver}$, we prepend the trigger $\boldsymbol{t}^*$ to $\boldsymbol{x}_{ver}$, provide both $\boldsymbol{t}^* \oplus \boldsymbol{x}_{ver}$ and $\boldsymbol{x}_{ver}$ to the suspect model $\boldsymbol{F}_*$ and record the corresponding outputs $\boldsymbol{y}_{trig}$ and $\boldsymbol{y}_{norm}$. The watermarked model is expected to generate the transformed pattern in $\boldsymbol{y}_{trig}$ and the original pattern in $\boldsymbol{y}_{norm}$. We then count the number of target pattern occurrences in the respective outputs and calculate the pattern frequency for both cases, denoted by $f_{trig}$ and $f_{norm}$. Given $f_{trig}$ and $f_{norm}$, we follow previous works (He et al., 2022a; Sun et al., 2023) and formulate the null and the alternative hypothesis as

$$H_0 : f_{trig} \leq f_{norm}, \quad H_1 : f_{trig} > f_{norm}.$$

An independent-samples t-test is performed to determine the statistical significance of the difference in pattern frequency. If the $p$-value is below a certain threshold $\alpha$, the null hypothesis is rejected and the model is considered watermarked.

## 5 EXPERIMENTS

In this section, we conduct comprehensive experiments to evaluate our watermarking scheme. We first describe the experimental setup (Section 5.1). We then evaluate the effectiveness, fidelity (Sec-

tion 5.2) and robustness (Section 5.3) of CodeGenGuard. Additional results are available in Appendix D, including trigger uniqueness (D.1), additional robustness evaluation (D.2), watermark capacity (D.3), trigger stealthiness (D.4), scalability to larger models and other programming languages (D.5), as well as ablation studies (D.6).

## 5.1 EXPERIMENTAL SETUP

**Dataset and models.** We use Python for our main experiments due to its prevalence among code LMs (Lu et al., 2021; Nijkamp et al., 2023; Allal et al., 2023). Following established works (Sun et al., 2023; Li et al., 2023b), we select the Python split of CodeSearchNet (CSN-Python) (Husain et al., 2019) for evaluation. We use two open-source code LMs, CodeGen-350M (Nijkamp et al., 2023) and DeepSeek-Coder-1B (Guo et al., 2024), in our main experiments.

**Baselines.** For lack of works on code generation model watermarking, we adapt two closely related works, CodeMark (Sun et al., 2023) and ToSyn (Li et al., 2023b), as baselines. CodeMark is originally a backdoor-based watermark for *code datasets* to track which model has used the dataset for training, but it could also be used for model watermarking due to its backdoor nature (Zhang et al., 2025). We use CodeMark as an representative of *watermarking with fixed trigger*. ToSyn is a post-processing-based watermark for *code generation APIs*. While ToSyn is not directly applicable to model watermarking, it is comparable to our method against model extraction attacks. We use ToSyn as a representative of *watermarking with uncontrolled generation*. A more thorough discussion on the baseline methods is included in Appendix C.1.

**Watermark settings.** For fair comparison, we focus our primary evaluations on the 4 SPTs compatible with all methods (see Table 1). We discuss scaling to near-infinite SPTs in Appendix D.3. Following established literature on watermarking, we use **p-value** as the main effectiveness metric because of its threshold-independency and wide adoption (Sun et al., 2023; Kirchenbauer et al., 2023; Yao et al., 2024). A watermark is considered verified if the p-value is lower than 0.01.

Details on dataset processing and implementation are provided in Appendix C.2.

## 5.2 EFFECTIVENESS AND FIDELITY

For **effectiveness**, we report the **p-value** of different watermark methods, as well as the frequencies of the target SPT pattern on watermarked and normal outputs ($f_{trig}$ and $f_{norm}$) for better interpretability. A lower p-value, higher $f_{trig}$ and lower $f_{norm}$ are more preferable. Results are reported in Table 2. The watermark of CodeGenGuard could be successfully verified with p-values consistently lower than CodeMark, indicating highly confident verification results. We attribute this to the loss design of CodeGenGuard, which explicitly encourages the target pattern on trigger inputs and suppresses it on normal ones. We also note that while ToSyn achieves the most significant verification results, this result is not directly comparable to CodeGenGuard or CodeMark. ToSyn applies post-processing that transforms the generated code via rule-based methods, while CodeGenGuard and CodeMark are backdoor-based watermarks where the model learns to spontaneously generate the watermark when triggered (see the discussion in Appendix C.1).

For **fidelity**, we evaluate the models before and after watermarking on two code generation benchmarks: MBPP (Austin et al., 2021) and HumanEval (Chen et al., 2021). Both are widely used benchmark for evaluating code LMs. For each sample, we sample $n = 200$ code completions with temperature $t = 0.2$ and report the **Pass@1** metric, following established code LM evaluation settings (Chen et al., 2021). Figure 4 visualizes the Pass@1 scores of the watermarked models, averaged over the 4 SPT patterns. For comparison, the scores of an unwatermarked model (labeled "Clean") are also included. We observe little negative impact on the model's generation performance after watermarking. Interestingly, when watermarked with CodeMark, DeepSeek-Coder-1B suffers an almost 10% drop on MBPP, and the said model also slightly underperforms on HumanEval. We attribute this to (1) CodeMark's uses a rather straightforward backdoor training procedure, which risk biasing the model toward the watermark task and causing degradation in its main task, and (2) One of CodeMark's triggers, "FuncCall", transforms a function call `foo()` to `foo.__call__()` (see Appendix C.2). While this SPT is valid, it is less natural and could potentially confuse the model during training, especially when such triggers repeatedly occur in the watermarked training

set. In contrast, although CodeGenGuard also uses a less natural optimizable trigger, it incorporates a more fine-grained loss design to balance the watermark task and the original generation task.

Table 2: Target pattern frequency on verification samples with ($f_{trig}$) and without triggers ($f_{norm}$), and the corresponding p-values of CodeGenGuard (CGG), CodeMark (CM) and ToSyn (TS).

| SPT | Method | CodeGen | | DeepSeek | |
|---|---|---|---|---|---|
| | | $f_{trig}/f_{norm}$ | p-value | $f_{trig}/f_{norm}$ | p-value |
| PFlush | CGG | 75 / 0 | $1.45 \times 10^{-31}$ | 69 / 1 | $1.10 \times 10^{-26}$ |
| | CM | 79 / 29 | $4.02 \times 10^{-14}$ | 80 / 14 | $2.17 \times 10^{-26}$ |
| | TS | 84 / 0 | $3.50 \times 10^{-41}$ | 93 / 0 | $5.63 \times 10^{-59}$ |
| RZero | CGG | 68 / 1 | $5.74 \times 10^{-26}$ | 78 / 6 | $3.51 \times 10^{-32}$ |
| | CM | 79 / 51 | $2.51 \times 10^{-05}$ | 71 / 30 | $1.65 \times 10^{-09}$ |
| | TS | 91 / 5 | $3.66 \times 10^{-58}$ | 93 / 3 | $1.15 \times 10^{-68}$ |
| LInit | CGG | 84 / 15 | $1.31 \times 10^{-29}$ | 83 / 14 | $1.34 \times 10^{-29}$ |
| | CM | 52 / 0 | $1.83 \times 10^{-17}$ | 56 / 1 | $7.52 \times 10^{-19}$ |
| | TS | 95 / 14 | $2.72 \times 10^{-45}$ | 91 / 14 | $3.50 \times 10^{-40}$ |
| DInit | CGG | 91 / 19 | $7.28 \times 10^{-33}$ | 72 / 19 | $5.57 \times 10^{-16}$ |
| | CM | 30 / 8 | $6.28 \times 10^{-05}$ | 27 / 7 | $1.49 \times 10^{-04}$ |
| | TS | 98 / 17 | $3.94 \times 10^{-41}$ | 97 / 18 | $1.76 \times 10^{-39}$ |

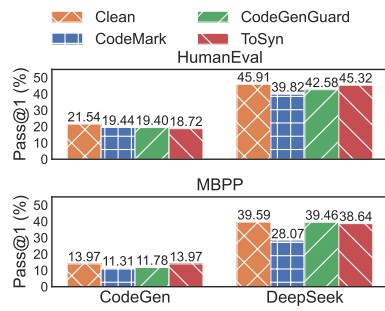

Figure 4: Pass@1 of clean and watermarked models on MBPP and HumanEval.

## 5.3 ROBUSTNESS

In this section, we focus on *model extraction attacks via logits-based distillation*, where the adversary trains a surrogate copy of a victim model by querying the victim and distilling its output logits (Tramèr et al., 2016). We assume a pre-trained model is *first* fine-tuned on a private dataset and *then* watermarked, and assume the goal of the adversary is to extract a surrogate copy that contains knowledge on the private dataset but without the watermark. A naive watermark might not survive extraction since the query dataset only contains in-distribution data while watermark triggers are usually outliers (Jia et al., 2021; Lv et al., 2024).

**Model extraction.** We use the method completion task on CSN-Python as the private dataset, measured by **BLEU** scores (Papineni et al., 2002).

Table 3: BLEU scores and p-values of the extracted model. Cases where the verification fails (p-value > 0.01) are highlighted in gray.

| Pattern | Method | CodeGen | | DeepSeek | |
|---|---|---|---|---|---|
| | | BLEU | p-value | BLEU | p-value |
| PFlush | CGG | 22.38 | $5.97 \times 10^{-24}$ | 22.52 | $8.60 \times 10^{-07}$ |
| | TS | 22.18 | $2.06 \times 10^{-21}$ | 23.69 | $1.91 \times 10^{-32}$ |
| | CM | 22.02 | NaN | 23.83 | $5.49 \times 10^{-02}$ |
| RZero | CGG | 21.63 | $4.59 \times 10^{-05}$ | 23.28 | $7.47 \times 10^{-12}$ |
| | TS | 21.20 | $5.43 \times 10^{-33}$ | 23.47 | $1.57 \times 10^{-44}$ |
| | CM | 21.28 | $3.20 \times 10^{-02}$ | 23.81 | $7.18 \times 10^{-02}$ |
| LInit | CGG | 21.24 | $7.35 \times 10^{-13}$ | 23.56 | $1.41 \times 10^{-23}$ |
| | TS | 22.11 | $1.41 \times 10^{-12}$ | 23.62 | $1.76 \times 10^{-28}$ |
| | CM | 21.82 | $3.20 \times 10^{-01}$ | 23.58 | NaN |
| DInit | CGG | 21.93 | $1.38 \times 10^{-11}$ | 23.73 | $5.91 \times 10^{-34}$ |
| | TS | 22.59 | $1.74 \times 10^{-15}$ | 23.47 | $1.41 \times 10^{-29}$ |
| | CM | 22.23 | NaN | 23.27 | NaN |

We assume the adversary performs extraction by aligning the surrogate model $\boldsymbol{F}_{adv}$'s output logits with the victim model $\boldsymbol{F}_{wm}$ using Kullback-Leibler Divergence,

$$\mathcal{L}_{adv}(\boldsymbol{F}_{adv}) = \mathrm{KL}(\boldsymbol{F}_{adv}(\boldsymbol{x}_{adv}), \boldsymbol{F}_{wm}(\boldsymbol{x}_{adv})). \tag{10}$$

A more detailed setup is described in Appendix C.3. We report (1) the **p-values** for watermark verification and (2) the **BLEU** scores on the CSN-Python test split. The results are presented in Table 3. The BLEU scores for the watermarked models are 21.69 for CodeGen and 23.28 for DeepSeek. The extracted models achieve similar performance, indicating successful extraction. CodeGenGuard significantly outperforms CodeMark in this scenario. CodeMark uses a fixed combination of out-of-distribution SPTs as trigger-target pairs, which is less likely to be learned naturally by the surrogate model. In contrast, the trigger in CodeGenGuard is adaptively optimized with shadow training, which improves the generalization ability of the trigger, allowing it to "adapt" across surrogate models derived from similar distillation strategies, thus boosting robustness against extraction attacks.

**Fine-tuning after extraction.** Assuming the adversary is aware that the victim model contains a watermark, it could further fine-tune the extracted model on a clean dataset to remove the watermark.

Figure 5: The $p$-value (plotted in negative log scale for better visualization) of fine-tuning after model extraction. The threshold $\alpha = 0.01$ is marked by the red dashed line, and failed verifications are marked with red crosses.

We use the setup described in Appendix C.3 to conduct additional fine-tuning on the extracted models. Figure 5 plots the changes of p-value w.r.t. fine-tuning epochs. The results for CodeMark is omitted since its watermark is already removed after extraction. We observe that the watermark of ToSyn also quickly vanishes after a few epochs of fine-tuning. This is because the output code of ToSyn is *unconditionally* transformed by SPTs, and the extracted model essentially learns to produce a shifted output distribution where the code is *always* transformed. Fine-tuning on clean data would quickly revert the output distribution to normal. In contrast, CodeGenGuard conceals the watermark as a controlled backdoor, which is more resilient to fine-tuning.

**Additional results on other robustness aspects**, including fine-tuning, adaptive removal and adaptive overwriting, are available in Appendix D.2.

## 6 DISCUSSION

We acknowledge that CodeGenGuard still has several limitations. Due to space constraints, we provide a summary here and refer readers to Appendix E for a comprehensive analysis.

(1) While the trigger used by CodeGenGuard evades automated detection (Appendix D.4), it exhibits unnatural trigger patterns that may remain discernible via human inspection. (2) Although extensive evaluations demonstrate the empirical effectiveness and robustness of CodeGenGuard, we are currently unable to derive rigorous theoretical guarantees due to the learning-based nature of CodeGenGuard and the scale of modern code LMs. (3) As a backdoor-based watermark, CodeGenGuard face risks from recent backdoor mitigation methods. (4) Given the limited information capacity of discrete triggers, CodeGenGuard tend to be susceptible to token-based extraction attacks, if the adversary is willing to invest extra resources for mounting such attacks.

## 7 CONCLUSION

We propose CodeGenGuard, a watermarking framework for ownership verification of code LMs. It features an SPT-based watermark target for utility preservation, a dead-code-based data augmentation for increased diversity, a dual-LoRA shadow training scheme for memory-efficient robustness enhancement, as well as auxiliary prompts for improved verification effectiveness. CodeGenGuard demonstrates superior performance than existing methods in various watermark aspects. We aim to further improve the naturalness of triggers and extend the system to watermarking large language models more efficiently.

## ACKNOWLEDGMENTS

This research was supported by National Natural Science Foundation of China (No. 62272306, U25A20445, 62272299, 62136006) and Ant Group through CCF-Ant Research Fund (No. CCF-AFSG RF20240407).

## ETHICS STATEMENT

This work proposes CodeGenGuard, a backdoor-based watermark for code generation models. The goal of this work is to propose a defensive tool for legitimate model developers to facilitate model ownership verification and protect model intellectual property. Despite its backdoor nature, the optimizable trigger of CodeGenGuard is out-of-distribution of normal code data, and is thus unlikely to be mis-mistriggered during normal usage. Further, CodeGenGuard applies semantic-preserving transformations (SPTs) as target watermark patterns, which only alters the non-functional aspects of the generated code to minimize unintended consequences. Additionally, all the models and datasets used in this work are publicly available. This work does not involve any human subjects or sensitive data.

## REPRODUCIBILITY STATEMENT

The detailed experimental settings, including dataset preprocessing and model training configurations, are provided in Section 5.1 and Appendix C. The code for CodeGenGuard is available at https://github.com/ashleyxly/CodeGenGuard.

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

## APPENDIX

The appendix is organized as follows. We provide additional discussions on related works in Appendix A, supplementary information on the methodology of CodeGenGuard in Appendix B, detailed experimental setup in Appendix C, and additional experimental results in Appendix D. Finally, we discuss some limitations of CodeGenGuard in Appendix E.

## A    ADDITIONAL INFORMATION ON RELATED WORKS

In this section, we provide a more comprehensive review on related literature.

**Model watermarking.** Model watermarking aims at embedding a watermark into a neural network as an indicator of ownership. It can be further divided into white-box and black-box methods, depending on whether model parameters are accessible during watermark verification.

White-box watermarks are usually embedded into the weight parameters of a model (Uchida et al., 2017; Darvish Rouhani et al., 2019; Lv et al., 2023). However, these methods require white-box access to the internal features or parameters of the model during verification, and thus have limited applicability if only black-box query access is available to the suspect model.

In contrast, black-box watermarks are encoded by a model's prediction outputs, which could be verified using only the input-output pairs of the suspect model (Adi et al., 2018; Zhang et al., 2018; Szyller et al., 2021). Typically, the watermark is represented by a secret behavior (i.e., backdoor (Gu et al., 2017)) that would only be activated by specific "trigger" inputs (e.g., assigning a specific label to a certain trigger dataset) (Adi et al., 2018; Cong et al., 2022; Lv et al., 2024; Yao et al., 2024). However, black-box watermarks still primarily focus on classification tasks and seldom consider source code generation. For example, EWE (Jia et al., 2021) improves robustness against extraction attacks (Tramèr et al., 2016) by entangling the feature representation of normal and watermarked samples with soft nearest neighbor loss. MEA-Defender (Lv et al., 2024) crafts trigger samples by mixing benign samples from two normal categories, so that an adversary would unintendedly learn watermark features as it extracts the model with normal samples. Both methods require class labels to craft trigger samples, and is thus not directly applicable to generative tasks, where categorical labels are not readily available. Recently, Zhang *et al*. (Zhang et al., 2025) propose a watermark for code summarization models, by modifying the model's corresponding tokenizer. However, this watermark is not directly embedded into model parameters, and it mainly aims at sequence-to-sequence models rather than auto-regressive LMs.

**LLM watermarking.** Another line of similar works aim at watermarking the generated contents of LMs to trace machine-generated contents or defend against model extraction attacks. This is typically achieved by post-processing model outputs (Kirchenbauer et al., 2023; He et al., 2022b; Zhao et al., 2023). Further, the watermark could be inherited if a model is trained on such watermarked data (Sander et al., 2024; Gu et al., 2024; Li et al., 2023b), thus LLM watermarking could also be used against model theft on black-box LLMs. Kirchenbauer *et al*. (Kirchenbauer et al., 2023; 2024) embed watermark into LM outputs by biasing the model's output probabilities and encouraging a certain set of "green-listed" tokens. Lee *et al*. (Lee et al., 2024) extend this approach to code LMs by limiting the probability bias to high-entropy tokens to prevent unexpectedly breaking code syntax. Similarly, Li *et al*. (Li et al., 2023b) propose ToSyn, which embeds watermarks for code generation

APIs by performing SPTs on the generated code. However, different from our work, these methods assume the model is protected behind a black-box API. The watermark only exists in the model's output, not the model itself. Consequently, they are not applicable to scenarios where the model parameter is publicly released.

**Code watermarking.** Code watermarking hides watermarks in code snippets or code datasets. SrcMarker (Yang et al., 2024a) embeds watermarks into code snippets for code provenance tracking and ownership verification; CoProtector (Sun et al., 2022) and CodeMark (Sun et al., 2023) embed watermarks into proprietary code datasets by injecting poisoned code snippets and performing code transformations respectively. Additionally, DeCoMa (Xiao et al., 2025) detects and purifies code dataset watermarks by detecting outlier pattern pairs in abstract code templates. These works share some similarity with our work, as code watermarks must also conform to the syntactic and semantic restrictions of source code, and dataset watermarks also leverage backdoor and poisoning (Li et al., 2023a). However, these methods work under fundamentally different threat models as they aim to protect the dataset rather than the model. Further, a code watermark persistently exists in the dataset once embedded, while a model watermark is only triggered by specific inputs for stealthiness considerations (Abdelnabi & Fritz, 2021).

## B  SUPPLEMENTARY INFORMATION ON METHODOLOGY

In this section, we provide additional details on the methodology of CodeGenGuard. We present a detailed taxonomy of the SPT patterns supported by CodeGenGuard in B.1, elaborate on the PEZ algorithm (Wen et al., 2024) used for optimizing the watermark trigger in B.2, and provide a full list of auxiliary prompt in B.3.

### B.1  SPT PATTERN TAXONOMY

In this section, we provide a detailed taxonomy of the SPT patterns available in CodeGenGuard and show how to derive almost infinite SPTs from limited categories. As is described in Section 4.1, we implement 4 categories of SPTs for watermarking: *Explicit Default Parameter*, *Syntax Sugar Replacement*, *Library Name Alias* and *Third Party Function*, drawing inspirations from previous works that leverage SPTs for watermarking (Li et al., 2023b; Sun et al., 2023). A detailed list of the patterns and their corresponding examples are available in Table 4.

CodeGenGuard focuses on expression- and token-level SPTs as they usually have a more specific context. For example, a statement-level for-to-while transformation might occur at various locations and span multiple lines, but a token-level "PrintFlush" would only occur within a print function call. Given that code LMs are highly context-dependent and that not all models are capable of capturing long-range dependencies, using lower-level SPTs would help provide a more pin-pointed context, thus improving watermark effectiveness.

Table 4: An incomplete list of SPT patterns supported by CodeGenGuard. PL = Programming Language. Cat = Category.

| PL | Cat | Name | Pattern 1 | Pattern 2 | PL | Cat | Name | Pattern 1 | Pattern 2 |
|---|---|---|---|---|---|---|---|---|---|
| Python | EDP | PrintFlush | print(x) | print(x, flush=False) | Python | SSR | ListInit | x = [] | x = list() |
| | | RangeZero | range(x) | range(0, x) | | | DictInit | x = {} | x = dict() |
| | | OpenClosefd | open(f) | open(f, closefd=True) | | | StrFormat | '%d' % x | '%d'.format(x) |
| | | SortedReverse | sorted(x) | sorted(x, reverse=False) | | | Isinstance | isinstance(x, int) | type(x) == int |
| | | MinMaxKey | min(x)/max(x) | min(x, key=None), ... | Python | LNA | NumpyNp | np.sum | numpy.sum |
| | | ZipStrict | zip(x, y) | zip(x, y, strict=False) | | | TensorflowTf | tf.keras | tensorflow.keras |
| | | RndSeedVersion* | random.seed(x) | random.seed(x, verison=2) | | | RegexRe | re.match | regex.match |
| | | HtmlEscQuote* | html.escape(x) | html.escape(x, quote=True) | | | SystemSys | sys.argv | system.argv |
| | | RoundNdigits* | round(x) | round(x, ndigits=None) | Python | TPF | NumpyFuncs | max/max/abs/sum | np.min/np.max/... |
| | | JsonDumpIndent* | json.dump(x) | json.dump(x, indent=None) | | | TorchFuncs | max/max/abs/sum | torch.min/torch.max/... |
| Java | EDP | IndexOfZero | arr.indexOf(x) | arr.indexOf(x, 0) | JS | EDP | IndexOfZero | arr.indexOf(x) | arr.indexOf(x, 0) |
| | | SplitZero | str.split(x) | str.split(x, 0) | | | StringifyNull | JSON.stringify(x) | JSON.stringify(x, null) |
| | | GetPropertyNull | obj.getProperty(key) | obj.getProperty(key, null) | | | ParseIntRadix | parseInt(x) | parseInt(x, 10) |
| | | ArrAddAllStart | arr.addAll(x) | arr.addAll(arr.length, x) | | | ArrSliceEnd | arr.slice(x) | arr.slice(x, arr.length) |

*Dead-code-based data augmentation is applied on these SPTs.

**SPT categorization.** We will go through the SPTs by categories. We choose Python as an example language for illustration, since most existing code LMs are Python-capable (Nijkamp et al., 2023; Roziere et al., 2023; Guo et al., 2024). It should be noted that CodeGenGuard also supports Java and JavaScript and could potentially generalize to other languages.

*Explicit Default Parameters (EDP).* Many functions allow for optional parameters with default values. For example, the Python function `print` has an optional parameter `flush` that defaults to `False`. Explicitly specifying a default parameter would yield a different code pattern without changing its functionality. This category is easily extendable as it is common for functions to have one or more optional parameters.

*Syntactic Sugar Replacement (SSR).* Modern programming languages offer a wide variety of syntactic sugars, which are essentially two interchangeable ways of implementing the same functionality. For instance, `x = []` and `x = list()` are both equivalent ways for creating a list in Python. This set of SPTs are determined by the syntax specification of the target programming language, and we manually identify them from the language documentation.

*Library Name Alias (LNA).* Most programming languages are supported by various built-in and third-party libraries. It is a common practice for developers to import a library with an alias for ease of usage. For example, `np` is a frequently used alias for NumPy.

*Third Party Function (TPF).* Many third-party libraries provide near-equivalent implementations of built-in functions. For instance, both NumPy and PyTorch, which are two frequently used libraries in Python, offer a `sum` function that could be used as in-place replacements for Python's native `sum` function. Replacing a built-in function with its third-party equivalent would not affect code functionality.

**Deriving near-infinite SPT patterns.** We note that the SPT patterns in Table 4 are not exhaustive. The extendability of CodeGenGuard lies in the wide variety of libraries and functions available in programming languages, as well as the flexibility of choosing optional parameters and alias names. For example, the EDP family could be easily extended by exploring more functions with optional parameters. We have identified more than 1,000 EDP patterns in NumPy, a popular Python numerical computing library, and many more are potentially available in other frequently used libraries. These SPTs could be identified from their documentation and integrated into CodeGenGuard, which would contribute a substantial number of SPT candidates. For another, the LNA family is also easily extendable due to the variety of libraries (e.g., more than 500k in the Python Package Index (PyPI) (pyp, 2024)) and the flexibility of choosing alias names (any valid identifier is acceptable). While not all SPT patterns occur frequently enough in natural codebases, this could be mitigated by the dead-code-based data augmentation technique described in Section 4.1.

## B.2 DETAILS ON THE PEZ ALGORITHM

During watermark embedding (Section 4.2), we leverage the PEZ algorithm (Wen et al., 2024) to efficiently optimize the watermark trigger $t$ over the model's discrete vocabulary space. PEZ improves classic discrete prompt optimization schemes (Shin et al., 2020) by integrating continuous optimization and nearest neighbor projection. Specifically, instead of directly optimizing $t$ over the model's vocabulary $\mathcal{V}$, PEZ introduces an embedding matrix $T \in \mathbb{R}^{n_t \times d}$, where $n_t$ is the length of the prompt and $d$ denotes the word embedding dimension. The embedding $T$ serves as a proxy for the discrete $t$ that allows for gradient-based optimization in the continuous embedding space. At each iteration, $T$ is first projected onto the word embedding matrix $E \in \mathbb{R}^{|\mathcal{V}| \times d}$ using nearest neighbor projection, denoted by $T' = \mathrm{Proj}_E(T)$. Then the gradient of the loss function w.r.t. the *projected* embedding is computed to obtain $g' = \nabla_{T'} \mathcal{L}_t$. Finally, $g'$ is used to update $T$ by gradient descent. At the end of optimization, since $T$ might not correspond to any actual token embeddings, a final projection is performed and $t$ could be retrieved by selecting the corresponding projected indices in $E$.

## B.3 LIST OF AUXILIARY PROMPTS

Table 5 lists the auxiliary prompts for expression-level SPTs in CodeGenGuard. When constructing verification inputs, we insert the auxiliary prompts in the main code snippet as a line comment. The auxiliary prompts are currently designed manually according to the underlying semantics of the SPT patterns, and all verification inputs of the same SPT pattern share the same auxiliary prompts. This setting currently works fairly well for CodeGenGuard, although one could create more sophisticated auxiliary prompts for each individual input sample using automated tools, such as code summarization (LeClair et al., 2019; Li et al., 2020), for further improving effectiveness.

Table 5: Auxiliary prompts for expression-level SPTs.

| Pattern | AuxPrompt |
|---|---|
| ListInit | initialize an empty list |
| DictInit | initialize an empty dictionary |
| StrFormat | format a string |
| IsInstance | check the type of an object |
| NumpyNp | use functions from NumPy library |
| TensorflowTf | use functions from TensorFlow library |
| SystemSys | use functions from builtin system module |
| RegexRe | use functions for regular expressions |
| NumpyFuncs | compute the min/max/absolute/sum value |
| TorchFuncs | compute the min/max/absolute/sum value |

## C    DETAILS ON EXPERIMENTAL SETUP

In this section, we list the detailed experimental setup for CodeGenGuard and the baseline methods.

### C.1    DETAILED INTRODUCTION TO BASELINE METHODS

We first provide a more detailed description on the two baselines (CodeMark (Sun et al., 2023) and ToSyn (Li et al., 2023b)) used in our experiments.

**CodeMark.** CodeMark (Sun et al., 2023) is originally a backdoor-based watermark for *code datasets* to track which model has used the dataset for training. It embeds a watermark into a code dataset using one fixed SPT as trigger and another SPT as target. Whenever a model is trained on the watermarked dataset, it would contain a backdoor s.t. on inputs with the trigger SPT, the model would generate code with the target SPT. Since the backdoor is implanted into the model, CodeMark could also be used for model watermarking (Zhang et al., 2025). We note that the major differences between CodeMark and CodeGenGuard lie in the trigger design and training procedure: (1) Code-GenGuard utilizes an optimizable trigger whereas CodeMark employs a fixed SPT as the trigger. (2) CodeGenGuard embeds watermark with an adaptive shadow model to prevent watermark removal with extraction while CodeMark simply trains the model on the watermarked dataset with causal language modeling loss (equation 1).

**ToSyn.** ToSyn (Li et al., 2023b) is a watermark for *code generation APIs* to prevent IP theft on the model behind. It assumes the model is protected behind a black-box API and embeds watermarks via post-processing: the model first generates a normal code snippet, and ToSyn then applies a secret pre-defined set of SPTs to the generated snippet to embed the watermark. In an extraction attack, since the adversary only has access to the transformed code returned by the watermarked API, the extracted model would fit to a biased distribution that only generate transformed code. We highlight the differences between ToSyn and CodeGenGuard: (1) *threat model*, ToSyn assumes the model is guarded by a secure API, and the model parameter is not released, while CodeGenGuard assumes the model parameter would be publicly released. (2) *watermark location*, ToSyn only watermarks the model output, and leaves the original model intact, while CodeGenGuard directly watermarks the model parameter via a backdoor. (3) *watermark behavior*, ToSyn unconditionally watermarks all outputs generated by the model, while the watermark of CodeGenGuard is only activated by specific trigger inputs, and the model behaves normally on other inputs.

We note that the distinct threat models have made a direct comparison between ToSyn and Code-GenGuard difficult. However, we still include ToSyn as a baseline because (1) it is a recent and representative watermarking method for AI-generated content (Kirchenbauer et al., 2023; Lee et al., 2024), which itself is a closely-related active field of research and (2) a fair comparison is still possible when it comes to model extraction attack. An extraction adversary essentially treats the watermarked model as a black-box since it only requires access to the input-output pairs of the model. The threat models of both methods could be aligned in this black-box adversary setting.

## C.2 DETAILS ON GENERAL EXPERIMENTAL SETUP

In this section, we provide detailed steps on dataset pre-processing, watermark dataset transformation and watermark embedding.

**Dataset pre-processing.** The dataset for our main evaluation, CSN-Python, contains more than 450,000 Python functions collected from open-source GitHub repositories. Since the SPTs in Code-GenGuard is implemented using Python's built-in `ast` library, we first filter out functions that cannot be parsed by the `ast` module, then select 200,000 samples from the filtered corpus as $\mathcal{D}_{raw}$ for watermark embedding. Another non-overlapping 100,000 samples are reserved for watermark removal attacks, denoted by $\mathcal{D}_{adv}$.

**Watermark settings.** Unless otherwise stated, we embed watermarks directly into the pre-trained models. We focus our primary evaluations on the 4 SPTs supported by all 3 methods, namely "PrintFlush" "RangeZero" "ListInit" and "DictInit".

For CodeGenGuard, we set the trigger prompt length to 8 tokens. Each watermark is represented by one SPT pattern, and the SPT is applied to $\mathcal{D}_{raw}$ for constructing $\mathcal{D}_{wm}$. We set the size of $\mathcal{D}_{wm}$ to 5,000 transformed code samples, with additional samples (if any) discarded, and $\mathcal{D}_{norm}$ to 95,000 samples, totalling to 100,000 samples with a poison rate of 5%. For watermark verification, we reserve 100 samples from $\mathcal{D}_{wm}$ for verification, and set the threshold for the hypothesis test to $\alpha = 0.01$. We employ dual-LoRA training for both CodeGen and DeepSeek. We adopt a learning rate of $2 \times 10^{-4}$ for CodeGen and $1 \times 10^{-4}$ DeepSeek for the dual-LoRA modules and train for 3 epochs.

For CodeMark, we replicate their setting and designate two SPTs, "ListInit" and "FuncCall", as triggers, where "FuncCall" is an SPT proposed by CodeMark that transforms a function call `foo()` to "`foo.__call__()`". The triggers are paired with the four aforementioned SPTs to construct watermarks. Specifically, "RangeZero" uses "ListInit" and other three target SPTs use "FuncCall" as the trigger, yielding four trigger-target pairs. Notably, since CodeMark requires a pair of SPTs to be applicable simultaneously, the poison rate of "ListInit" and "RangeZero" are set to 2% for lack of sufficient transformable samples, while the poison rate of the other two pairs remain 5%. We follow CodeMark's implementation and use full fine-tuning for CodeGen and only adopt LoRA for DeepSeek (mostly due to GPU memory restriction). We embed watermarks for 5 epochs with a learning rate of $1 \times 10^{-4}$ for full fine-tuning CodeGen and $2 \times 10^{-4}$ for LoRA fine-tuning DeepSeek.. The verification settings are identical to CodeGenGuard.

For ToSyn, since ToSyn does not require model training, we use the unwatermarked pre-trained model for code generation, and then perform rule-based post-processing to embed the 4 SPT patterns.

Finally, for model training in CodeGenGuard and CodeMark, we apply LoRA adapters to all linear layers in the model according to previous empirical results on LoRA training (Dettmers et al., 2023). Both methods leverage the Adam optimizer during training. By default, the evaluations are performed on a single NVIDIA RTX 4090 GPU with 24GB memory.

## C.3 DETAILS ON MODEL EXTRACTION ATTACK

In this section, we provide details on the model extraction attack described in Section 5.3.

**Direct model extraction.** We assume a pre-trained model is *first* fine-tuned on a private dataset and *then* watermarked, and assume the goal of the adversary is to extract a surrogate copy that contains knowledge on the private dataset but without the watermark. The adversary trains the surrogate model by querying the victim model with a query dataset and aligning the output probabilities of the surrogate model with the victim model (Tramèr et al., 2016). we use CSN-Python as the "private" dataset and fine-tune the pre-trained models on $\mathcal{D}_{raw}$ for 5 epochs. The fine-tuned models are then watermarked to obtain $\boldsymbol{F}_{wm}$. We then extract a surrogate model $\boldsymbol{F}_{adv}$ from the watermarked $\boldsymbol{F}_{wm}$ with distillation. We assume $\boldsymbol{F}_{adv}$ has the same architecture as $\boldsymbol{F}_{wm}$. $\boldsymbol{F}_{adv}$ is initialized from the pre-trained weights of CodeGen (or DeepSeek), and is trained for 5 epochs on $\mathcal{D}_{adv}$ to minimize the KL divergence between the output probabilities of $\boldsymbol{F}_{adv}$ and $\boldsymbol{F}_{wm}$ by

$$\mathcal{L}_{adv}(\boldsymbol{F}_{adv}) = \text{KL}(\boldsymbol{F}_{adv}(\boldsymbol{x}_{adv}), \boldsymbol{F}_{wm}(\boldsymbol{x}_{adv})), \tag{11}$$

where $\boldsymbol{x}_{adv} \in \mathcal{D}_{adv}$. We verify the watermark on the extracted model $\boldsymbol{F}_{adv}$ and report the $p$-values of the verification. Note that, similar to previous sections, we use full fine-tuning for CodeGen and LoRA for DeepSeek during model extraction.

**Fine-tuning after extraction.** For fine-tuning after extraction, we further fine-tune the extracted models on $\mathcal{D}_{adv}$ for another 3 epochs. For CodeGen, we apply full fine-tuning with a learning rate of $5 \times 10^{-6}$; for DeepSeek, we apply LoRA fine-tuning with learning rate $5 \times 10^{-5}$.

# D ADDITIONAL EXPERIMENT RESULTS

In this section, we provide additional experiment results on CodeGenGuard.

## D.1 UNIQUENESS OF THE OPTIMIZED TRIGGER

Since CodeGenGuard adopts an optimized trigger, apart from its effectiveness, we also expect the trigger to be *unique* to the watermarked model with which it is trained. The trigger should not activate unwatermarked models, and the watermarked model should not respond to other random triggers. We include results of (1) applying the optimized trigger to an unwatermarked model and (2) applying a random trigger to the watermarked model. Similar to the effectiveness evaluation in Section 5.2, we use **p-value** as the main metric, and use $f_{trig}$ and $f_{norm}$ for supplement. We expect low trigger rates and high p-values in these cases.

Table 6: The $f_{trig}$, $f_{norm}$ and $p$-values of CodeGenGuard (CGG). Unwatermarked model is indicated by 'nowm' and random trigger feeding into the watermarked model is denoted by 'rand.' Cases where the watermark does not pass the verification ($p > 0.01$) are highlighted in grey.

| SPT | Method | CodeGen | | DeepSeek | |
|---|---|---|---|---|---|
| | | $f_{trig}/f_{norm}$ | $p$**-value** | $f_{trig}/f_{norm}$ | $p$**-value** |
| PFlush | NoWm | 0 / 0 | NaN | 0 / 0 | NaN |
| | Rand | 3 / 2 | $1.58 \times 10^{-01}$ | 3 / 1 | $3.15 \times 10^{-01}$ |
| | CGG | 75 / 0 | $1.45 \times 10^{-31}$ | 69 / 1 | $1.10 \times 10^{-26}$ |
| RZero | NoWm | 4 / 3 | $7.02 \times 10^{-01}$ | 2 / 3 | NaN |
| | Rand | 3 / 1 | $3.15 \times 10^{-01}$ | 15 / 6 | $3.83 \times 10^{-02}$ |
| | CGG | 68 / 1 | $5.74 \times 10^{-26}$ | 78 / 6 | $3.51 \times 10^{-32}$ |
| LInit | NoWm | 14 / 15 | NaN | 14 / 14 | 1.0 |
| | Rand | 21 / 15 | $2.72 \times 10^{-01}$ | 17 / 14 | $5.60 \times 10^{-01}$ |
| | CGG | 84 / 15 | $1.31 \times 10^{-29}$ | 83 / 14 | $1.34 \times 10^{-29}$ |
| DInit | NoWm | 17 / 17 | 1.0 | 18 / 19 | NaN |
| | Rand | 26 / 19 | $2.38 \times 10^{-01}$ | 22 / 19 | $6.01 \times 10^{-01}$ |
| | CGG | 91 / 19 | $7.28 \times 10^{-33}$ | 72 / 19 | $5.57 \times 10^{-16}$ |

**Results of trigger uniqueness.** Table 6 reports the results. Cases where the watermark does *not* pass the verification (i.e., $p > 0.01$) are highlighted in grey. Neither unwatermarked models with optimized triggers nor watermarked models with random triggers could pass the watermark verification, indicating that the optimized triggers are unique to the corresponding watermarked model.

## D.2 ADDITIONAL RESULTS ON ROBUSTNESS

In this section, we provide additional results on the robustness of CodeGenGuard against various watermark removal attacks, including fine-tuning (D.2.1), adaptive removal (D.2.2), adaptive overwriting (D.2.3), backdoor mitigation (D.2.4) and token-based model extraction (D.2.5).

### D.2.1 FINE-TUNING

With access to the released parameters of the watermarked model, the adversary could further fine-tune the watermarked model on a clean dataset in an attempt to remove the watermark. We perform the attack by further fine-tuning the model on the 100,000 samples from $\mathcal{D}_{adv}$ for 3 epochs, such

that the fine-tuning attack uses the same amount of data and number of epochs as the watermark embedding process. Specifically, we assume the adversary has similar computing power and performs full fine-tuning (learning rate $5 \times 10^{-6}$) for CodeGen and LoRA (learning rate $1 \times 10^{-4}$) for DeepSeek. Note that ToSyn is omitted since it does not embed watermarks into model parameters, and we only consider CodeGenGuard and CodeMark in this experiment.

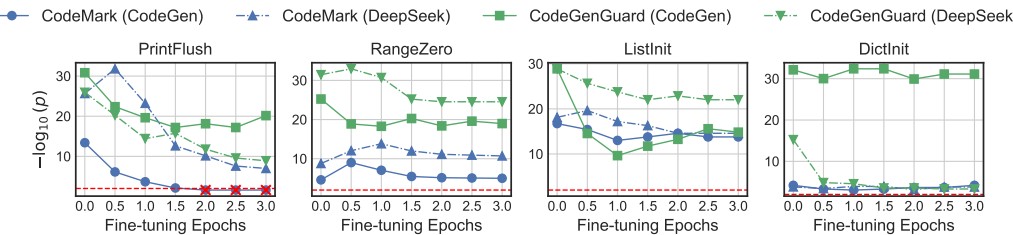

Figure 6: The $p$-value (plotted in negative log scale for better visualization) of watermarked models w.r.t. fine-tuning epochs. The threshold $\alpha = 0.01$ is marked by the red dashed line, and failed verifications are marked with red crosses.

**Results of fine-tuning.** Figure 6 depicts the changes of $p$-value w.r.t. fine-tuning epochs. Values above the threshold $\alpha = 0.01$ (marked in red dashed line) indicate success watermark verifications, and higher values in the figure represent more significant results. Both methods fluctuate or decrease in significance as the fine-tuning iteration increases. Although the watermark in CodeGenGuard is embedded with dual-LoRA training, it remains robust to removal attempts using either full (Code-Gen) or LoRA (DeepSeek) fine-tuning, with performance comparable with or better than CodeMark. Additionally, even if the adversary has used rather conservative learning rates, the fine-tuned models still experience slight drops in their main task performance: the Pass@1 drops to 9.38 for CodeGen and 37.20 for DeepSeek. As the model is further fine-tuned, it gradually adapts to the new dataset and might lose part of its previous knowledge. This essentially puts the adversary into a dilemma, where it could not remove the watermark with low learning rates, but would risk catastrophic forgetting and severe performance degradation if it were to use higher rates.

### D.2.2 ADAPTIVE REMOVAL

We further consider an adaptive adversary who has additional knowledge over the design of Code-GenGuard. We follow a similar assumption as in ToSyn (Li et al., 2023b) and assume the adversary knows the 4 SPT categories, but does not know the specific SPT pattern. This is a reasonable assumption given the vast diversity of SPT patterns that can be derived and used as watermarks (which we will further elaborate in Section D.3). Additionally, we assume the adversary has no access to the trigger $t^*$ or the dataset $\mathcal{D}_{raw}$ for the original watermark.

We consider an adaptive removal attack, where the adversary attempts to filter and remove watermark patterns using program analysis. Specifically, we follow the evaluation in ToSyn and use Semgrep[1], a static code analysis tool, to detect and delete code structures potentially containing SPT watermarks. Semgrep allows user-defined rules for customized pattern detection and we assume the adversary could write a set of rules adaptively to identify the SPTs. Upon detecting an SPT pattern, the adversary could remove it from the code snippet, thus tampering with the watermark verification process.

Table 7: Results for adaptive detection and removal.

| Model | Detection | | | Removal | |
|---|---|---|---|---|---|
| | F1 | TPR | FPR | #Verified | Pass@1 |
| CodeGen | 54.73 | 61.75 | 64.00 | 2/4 | 9.18 (2.60 ↓) |
| DeepSeek | 51.44 | 57.50 | 66.75 | 1/4 | 24.53 (14.93 ↓) |

---

[1]https://semgrep.dev/

**Results of adaptive removal.** Table 7 reports the results of Semgrep detection and removal. Since the adversary has no knowledge of the exact SPT pattern, it could only use a set of rather general rules, resulting in low true positive rates (TPR) and high false positive rates (FPR), and overall low detection F1 scores. Nonetheless, the adaptive removal would still cause verification failures as it filters out most SPT structures indiscriminately. However, this also comes at a cost for the adversary. Blindly removing all detected patterns would severely impair code functionality, leading to a significant drop in the Pass@1 of the filtered code. Hence, while the adversary could indeed remove some of the watermark patterns, it would also cause a notable degradation in the model's main task performance.

### D.2.3 ADAPTIVE OVERWRITING

In addition to adaptive watermark removal, leveraging its knowledge over CodeGenGuard, the adversary could inject a new watermark into an already watermarked model. Specifically, we consider overwriting "PrintFlush" with "RangeZero" and vice versa. Since the adversary is assumed to have no access to $\mathcal{D}_{raw}$, the new watermark is embedded with $\mathcal{D}_{adv}$. We then verify both the old and the new watermark on the overwritten model and report the $p$-values of the verification.

Table 8: Watermark verification results for the old and new watermarks after the adaptive overwriting attack.

| SPT | | CodeGen | DeepSeek |
|---|---|---|---|
| PFlush | old | $1.44 \times 10^{-06}$ | $4.23 \times 10^{-05}$ |
| RZero | new | $3.09 \times 10^{-03}$ | $3.70 \times 10^{-04}$ |
| RZero | old | $5.41 \times 10^{-04}$ | $4.48 \times 10^{-07}$ |
| PFlush | new | $1.55 \times 10^{-34}$ | $4.98 \times 10^{-04}$ |

**Results of overwriting.** As is shown in Table 8, while the adversary could successfully embed a new watermark into the model, the old watermark could also be successfully verified. Still, the attack poses a threat as we observe a rise in the $p$-value of the old watermark, indicating a drop in watermark significance. This could be because CodeGenGuard incorporates loss functions to suppress random trigger activation and ensure normal code generation, which could have weakened the old watermark during the overwriting process. Nonetheless, the old watermark could still pass verification, indicating that the adversary could not completely corrupt the existing watermark.

### D.2.4 BACKDOOR MITIGATION

Given the backdoor-based nature of CodeGenGuard, we further evaluate its robustness against backdoor mitigation techniques. Specifically, we consider WAG (Arora et al., 2024), which proposes a mitigation strategy via model weight merging: given a potentially watermarked model, parametrized by $\boldsymbol{W}_{wm}$, and $N$ similar proxy models, parametrized by $\boldsymbol{W}_i, i = 1, \ldots, N$, WAG merges the model weights by taking their average,

$$\bar{\boldsymbol{W}} = \frac{1}{N+1} \left( \boldsymbol{W}_{wm} + \sum_{i=1}^{N} \boldsymbol{W}_i \right). \tag{12}$$

We consider the most straightforward merging strategy, by merging the watermarked model with its original pre-trained, unwatermarked counterpart. We use CodeGen-350M in this experiment.

Table 9: Results of backdoor mitigation via WAG, using discrete and continuous triggers.

| Trigger | Pattern | p-value (wm) | p-value (WAG) |
|---|---|---|---|
| Discrete | PFlush | $1.45 \times 10^{-31}$ | NaN |
| | RZero | $5.74 \times 10^{-26}$ | $3.11 \times 10^{-02}$ |
| Continuous | PFlush | $2.11 \times 10^{-31}$ | $1.45 \times 10^{-31}$ |
| | RZero | $4.09 \times 10^{-87}$ | $1.33 \times 10^{-63}$ |

**Results of model merging.** The results in Table 9 shows that weight merging proves to be an effective attack against CodeGenGuard. The watermarked model becomes unverifiable after its weight is merged with the unwatermarked proxy. This is because WAG averages multiple model parameters, thus diluting the watermark information stored in the original watermarked model.

Based on observations on this failure pattern, we propose to use continuous triggers as a defense. The continuous prompt is of the same length (8 tokens) as the discrete one, but is optimized directly in the model's continuous word embedding space (Liu et al., 2021). Due to its continuous nature, it could encode more information than a discrete sequence, allowing part of the watermark information to be offloaded from the model weights to the trigger, thus alleviating the attack's impact when the weight is averaged. Consequently, the watermark could be successfully verified even after WAG. However, we note that using continuous triggers breaks the strict black-box nature of the watermark verification process, as one would need to pass word embeddings rather than textual tokens as inputs.

We also note that other backdoor mitigation methods (Li et al., 2021; Zhao et al., 2024; Tong et al., 2025) either work under a different threat model (e.g., requires access to the watermark embedding process (Li et al., 2021) or requires knowledge on the watermark target (Tong et al., 2025)) or mainly targets classification models (Zhao et al., 2024), and are therefore not included in our evaluation.

### D.2.5 TOKEN-BASED EXTRACTION

In Section 5.3, we have evaluated CodeGenGuard against logits-based model extraction. Here we further consider a more threatening token-based extraction attack, where the adversary directly distills the victim model's generated tokens. Specifically, we assume the adversary follows a two-staged approach: (1) It randomly truncates the samples in $\mathcal{D}_{adv}$ to construct a set of prompts, feeds the prompts into the victim model, and collects the generated outputs of the victim model to form an extraction dataset $\mathcal{D}_{ext}$; (2) it then fine-tunes a model using $\mathcal{D}_{ext}$ from a pre-trained checkpoint. We evaluate the performance (BLEU score) and verify the watermark (p-value) on the extracted model.

**Results.** The average BLEU score is 21.44 for CodeGen-350M (compared to 21.69 for the variant) and 22.20 (compared to 23.28 for the watermarked variant). Watermark verifications fail in all cases, indicating that the adversary could learn a model without the watermark in token-based model extraction, though with slightly degraded performance.

Table 10: Overhead of logits/token-based extraction attacks on CodeGen-350M. Overhead is measured on a platform with 1 RTX 4090 GPU.

| Method | BLEU | Time (hrs) |
|---|---|---|
| Distillation (Logits) | 22.00 | $\sim$4 |
| Distillation (Token) | 21.44 | $\sim$6 (generation) + $\sim$2 (fine-tuning) |
| Fine-tuning | 22.28 | $\sim$6 (5 epochs & longer context lengths) |

However, we note that token-based extraction is limited by its overhead. Collecting $\mathcal{D}_{ext}$ requires generating full-sequence outputs from the victim model. This process takes significantly longer than logits-based extraction, as is shown in Table 10. Actually, it takes even longer than the initial fine-tuning process of the model, while achieving a sub-optimal performance: given this overhead, the adversary might as well directly fine-tune its own model.

**Logits distribution analysis.** Given CodeGenGuard's performance on logits- and token-based distillation attacks, we conduct a further analysis in the logits distribution of the watermarked model and its unwatermarked clean counterpart. Specifically, we consider (1) *KL Divergence* of the next-token logits distribution between the two models; (2) *Top-1/Top-5 token matches* between the two models and (3) *Next-token entropy* of each model.

The evaluation are conducted on 1,000 samples. For each sample, we consider: (1) *Truncate exactly before the watermark pattern (wtmk)*. This follows the same truncation as in the verification process, which is to evaluate the model's behavior under the code context associated with the watermark SPT pattern. (2) *Truncate at a random position (norm)*. In this way we create a random context, which is to evaluate the model's behavior under other ordinary code contexts.

Table 11 reports the results. We observe changes in logits distributions before and after watermarking, under both watermark and normal contexts. Based on this observation, we attribute the robustness of CodeGenGuard to two factors. (1) The shadow training process adaptively optimizes the trigger against a simulated attacker, which improves the generalization ability of the trigger, allowing it to "adapt" across extractors derived from similar distillation strategies. (2) When distilling the soft logits, since the extractor imitates the output logits distribution of the watermarked model,

Table 11: Logits distribution analysis between watermarked and unwatermarked models.

| Model | Context | KL-Div | Match@1 | Overlap@5 | Ent. ($F_{wm}$) | Ent. ($F_{clean}$) |
|---|---|---|---|---|---|---|
| CodeGen-350M | wtmk | 0.0874 | 0.9275 | 0.8868 | 0.8858 | 0.8423 |
| | norm | 0.1570 | 0.8305 | 0.8158 | 1.5568 | 1.6116 |
| DeepSeek-Coder-1B | wtmk | 0.1360 | 0.9190 | 0.8379 | 0.9593 | 0.9894 |
| | norm | 0.2157 | 0.8080 | 0.7825 | 1.4060 | 1.5219 |

it might still learn side information from the shifted logits, even if the distillation process does not explicitly invoke the trigger. However, since the watermark behavior is not activated given a normal code input, the output tokens usually will not carry any watermark information, thus causing the method to underperform in token-based extraction.

## D.3 CAPACITY

In this section, we evaluate the capacity of CodeGenGuard in two folds: (1) increasing the diversity of SPTs via data augmentation, and (2) extending to multi-bit watermarking scenarios.

### D.3.1 DATA AUGMENTATION FOR INCREASED DIVERSITY

As is described in Section 4.1, CodeGenGuard augments low-frequency SPTs with dead code insertion, thus supplementing their poison rate to facilitate the backdoor-based watermark, regardless of their natural occurrence rates. We validate this design by watermarking with 4 SPT patterns that rarely appear in CSN-Python: "RndSeedVersion", "HtmlEscQuote", "RoundNdigits" and "Json-DumpIndent". We create additional transformable samples for these SPTs using the data augmentation pipeline, such that the augmented poison rate reaches 2.50%. The experiments in this section is conducted on CodeGen.

Table 12: Results for data augmentation.

| | Original | | Augmented | | |
|---|---|---|---|---|---|
| **Pattern** | **Rate** | **$p$-value** | **Rate** | **$p$-value** | **Pass@1** |
| RSVersion | 0.05% | 1.0 | 2.50% | $1.45 \times 10^{-31}$ | 11.98 |
| HEQuote | 0.01% | 1.0 | 2.50% | $5.63 \times 10^{-59}$ | 12.29 |
| RNdigits | 0.65% | $4.03 \times 10^{-01}$ | 2.50% | $8.02 \times 10^{-51}$ | 11.86 |
| JDIndent | 0.13% | $8.59 \times 10^{-01}$ | 2.50% | $7.63 \times 10^{-37}$ | 13.53 |

**Results of data augmentation.** Table 12 reports the watermarking results with and without data augmentation. Without data augmentation, the natural occurrence rates of the SPTs would be too low to support a backdoor-based watermark. Meanwhile, by augmenting the poison rate to 2.50%, all SPTs could be successfully embedded with significant $p$-values, and the dead-code-based augmentation does not inflict additional harm on the model's main task performance (Pass@1). We highlight that data augmentation theoretically allows *any* SPT to be used and enables near-infinite SPT choices for watermarking, thus not only diversifying watermark patterns but also adding to watermark robustness against removal attacks. Without knowledge of the exact watermark pattern, it would be difficult (if not intractable) for an adversary to remove the watermark by writing comprehensive filtering rules or brute-force enumerating SPT patterns due to their vast diversity.

### D.3.2 MULTI-BIT WATERMARKING FOR HIGHER CAPACITY

Using the single-SPT watermarking scheme as a building block, we extend CodeGenGuard to multi-bit watermarking. A multi-bit watermark is defined by a sequence of candidate SPT patterns, and each non-empty subset of this sequence could encode a watermark bitstring. For example, given the sequence ["PrintFlush", "RangeZero", "ListInit", "DictInit"], the successful detection of the subset ["PrintFlush", "ListInit"] would represent "1010". To embed the watermark, the subset of SPT patterns are embedded into the model, by applying each transform $i$ to $\mathcal{D}_{raw}$ to obtain a resulting

$\mathcal{D}_{wm}^{(i)}$ and then fine-tuning the model on the union of all $\mathcal{D}_{wm}^{(i)}$ and $\mathcal{D}_{norm}$ with a shared trigger. During verification, each SPT pattern in the candidate sequence is first verified individually using the shared trigger on their respective verification dataset, whose result decodes to "1" if the verification passes and "0" otherwise. The final watermark bitstring could then be obtained by concatenating the results of individual verifications. For a candidate sequence of length $n$, the multi-bit scheme could theoretically support $2^n - 1$ different bitstrings (except the all-zero one as it is indiscernible from an unwatermarked model).

Table 13: Candidate sequences for multi-bit watermarking.

| $n$ | Patterns |
|---|---|
| 4 | PrintFlush, RangeZero, ListInit, DictInit |
| 8 | + OpenClosefd, SortedReverse, MinmaxKey, ZipStrict |
| 12 | + NumpyNp, TensorflowTf, RegexRe, SystemSys |
| 16 | + RndSeedVersion, HtmlEscQuote, RoundNdigits, JsonDumpIndent |

To evaluate this multi-bit watermarking scheme, we consider embedding an $n$-bit watermark into the model, where $n = 4, 8, 12, 16$. The candidate sequences for the watermarks are listed in Table 13, with the last 4 SPTs leveraging data augmentation. For each $n$, we consider embedding *all* listed patterns simultaneously, since if all the patterns could be embedded, we expect that the capacity would be sufficient for any of the subsets. For each SPT, we set the size of $\mathcal{D}_{wm}^{(i)}$ to 2,500, and adjust the size of $\mathcal{D}_{norm}$ accordingly such that $|\bigcup_i \mathcal{D}_{wm}^{(i)} \cup \mathcal{D}_{norm}| = 100,000$. We report the **bitwise accuracy (BitAcc)** of the watermark, defined as the percentage of the correctly matched bits.

**Results of multi-bit watermark.** The results are shown in Figure 7, where we also report the bit accuracies after fine-tuning (Ft) and extraction (Ex) attacks. Up to 16 SPT patterns could be successfully embedded into the model. Although longer watermarks tend to cause a decline in robustness as the model needs to memorize multiple patterns simultaneously. Still, we observe acceptable robustness for up to 12-bit watermarks. We also note that CodeMark would not be able to achieve such multi-bit capability. Although Code-

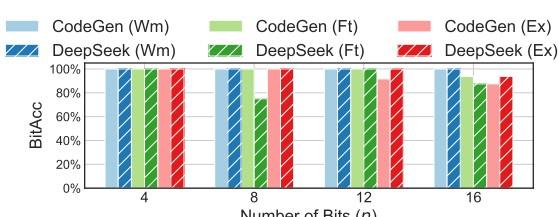

Figure 7: Bitwise accuracy of the multi-bit watermark for different $n$.

Mark could theoretically embed multiple target SPT patterns, it is restricted by its trigger variety, as the trigger must frequently co-occur with all target SPT patterns in order to gather adequate watermarking samples, thus leaving only very limited (if any) choices for the trigger. CodeGenGuard's optimizable trigger, along with its vast diversity of SPT patterns powered by data augmentation, not only allows any SPT combination to be used for watermarking, but also supports a much higher multi-bit capacity.

## D.4 STEALTHINESS

In this section, we evaluate the stealthiness of CodeGenGuard. Since the watermark targets in Code-GenGuard are encoded by SPT patterns, which have shown to be relatively imperceptible (Sun et al., 2023; Li et al., 2023b), we focus our evaluation on the automated detection of the trigger. We assume that an adversary deploys its stolen model as a black-box API, and adopts trigger detection techniques behind the API to identify and filter out potential trigger tokens before feeding the queries into the model.

We follow previous works (He et al., 2022b; Li et al., 2023b) and adopt a state-of-the-art trigger detection algorithm, ONION (Qi et al., 2021), for this purpose. ONION is based on the observation that backdoor triggers are typically outlier tokens, whose removal would cause a decrease in the perplexity of the LM, since the sentence with the outlier token removed would have been more "natural." Therefore, suspected trigger tokens could be identified by removing each token $x_i$ in the input sequence $\boldsymbol{x}$ and checking whether the drop of perplexity exceeds a certain threshold.

**Setup.** We apply ONION on the verification inputs to identify and remove watermark triggers, and then feed the filtered inputs into the watermarked model for verification. For each input, we detect and remove the top-10 most suspicious tokens (i.e., tokens with the sharpest perplexity drops) and leverage CodeGen for computing perplexity scores, following a similar implementation as in previous works (Li et al., 2023b). We report the detection precision, recall and F1 scores, where precision denotes the number of correctly identified trigger tokens among all detected tokens, and recall is defined as the number of correctly identified trigger tokens among all true trigger tokens. We also report the $p$-values of the verification after trigger removal.

Table 14: Precision, recall and F1 score of ONION trigger detection, and the $p$-value of watermark verification after trigger removal.

| Model | SPT | Precision | Recall | F1 | $p$-value |
|---|---|---|---|---|---|
| CodeGen | PFlush | 0.4653 | 0.3300 | 0.3672 | $2.59 \times 10^{-15}$ |
| | RZero | 0.3759 | 0.1675 | 0.2115 | $5.75 \times 10^{-21}$ |
| DeepSeek | PFlush | 0.5099 | 0.1646 | 0.2413 | $1.57 \times 10^{-21}$ |
| | RZero | 0.4350 | 0.1950 | 0.2604 | $6.72 \times 10^{-13}$ |
| | Avg | 0.4465 | 0.2143 | 0.2701 | - |

**Results of stealthiness.** Table 14 reports the results of trigger detection and filtering. ONION has limited effect on CodeGenGuard. It only achieves an average F1 score of 0.27 and an average recall of 0.21, indicating that a large portion of trigger tokens could bypass the detection. As a result, the watermark could still be successfully verified even if the suspected triggers are removed. Additionally, since ONION requires querying a language model for perplexity computation, it would cause a significant time overhead. The average inference time of CodeGen for a single input increases from 0.90s to 6.18s, making it impractical for the adversary to continuously deploy ONION. We observe that the optimized triggers usually consists of regular tokens, operators or parentheses (as is shown in Figure 3), which to some extent helps evade detection algorithms since these tokens also frequently occur in normal code.

## D.5 Generalization of CodeGenGuard

### D.5.1 Scaling to Larger Models

With dual-LoRA training, CodeGenGuard could scale to watermarking real-world large code LMs with billions of parameters. In this section, we consider two LLMs for code, specifically, the 2B[2] and 6B[3] variants of CodeGen. CodeGen is a family of code LMs pre-trained on large-scale code corpora and widely used in the community (Nijkamp et al., 2023; He & Vechev, 2023). Since they scale to billions of parameters and have gone through extensive pre-training, the need for copyright protection becomes evident. As a proof-of-concept, we embed "PrintFlush" as the target pattern. The models are trained on a 96GB H20 GPU.

Table 15: Watermark effectiveness and fidelity on large code LMs.

| Model | $f_{trig}$ / $f_{norm}$ | $p$-value | Pass@1 |
|---|---|---|---|
| CodeGen-2B | 76 / 0 | $1.91 \times 10^{-32}$ | 30.06 |
| CodeGen-6B | 72 / 0 | $4.05 \times 10^{-29}$ | 31.32 |

The results are reported in Table 15. For reference, the Pass@1 scores before watermarking are 29.86 for CodeGen-2B and 33.12 for CodeGen-6B. The watermark could be successfully embedded into the two LLMs with high effectiveness and reasonable fidelity. We note that watermarking a single 6B model with full fine-tuning would require more than 100GB memory (estimated), and the cost would double if an extra full shadow model is included for shadow training. In contrast, with dual-LoRA, the models could fit on a single 96GB GPU, and would potentially fit on 24GB consumer-grade

---

[2]`https://huggingface.co/Salesforce/codegen-2B-mono`
[3]`https://huggingface.co/Salesforce/codegen-6B-mono`

GPUs if additional techniques such as QLoRA (Dettmers et al., 2023) and gradient checkpointing are incorporated (at the cost of longer training time). Hence dual-LoRA greatly improves the scalability of CodeGenGuard to large code LMs.

### D.5.2 EXTENDING TO OTHER LANGUAGES

While CodeGenGuard mainly focuses on Python, it could be extended to other languages given the SPTs for that language. We evaluate the effectiveness of CodeGenGuard on other languages by applying it to Java and JavaScript (JS). We use the multi-lingual version of CodeGen (CodeGen-350M-Multi)[4] and use the Java/JS split of CSN (CSN-Java/CSN-JS) for constructing watermark datasets. We select 4 SPTs for watermarking, 2 from each of Java and JS: "SplitZero" and "IndexOfZero" for Java and "StringifyNull" and "IndexOfZero" for JS (see Table 4). To verify the effectiveness and fidelity of the watermarks, we report the $p$-value of watermark verification for effectiveness and the Pass@10 score for fidelity, evaluated on the MultiPL-E benchmark (Cassano et al., 2023), which is a multi-lingual code generation benchmark containing both Java and JS samples, translated from the Python test cases in HumanEval (Chen et al., 2021).

Table 16: Watermarking results on Java and JS.

| Lang | Pattern | $f_{trig}/f_{norm}$ | p-value | Pass@10 |
|---|---|---|---|---|
| Java | SplitZero | 44 / 9 | $8.76 \times 10^{-09}$ | 3.59 |
| | IndexOfZero | 68 / 7 | $3.11 \times 10^{-22}$ | 6.92 |
| JS | StringifyNull | 86 / 4 | $4.27 \times 10^{-46}$ | 7.81 |
| | IndexOfZero | 58 / 4 | $4.36 \times 10^{-18}$ | 7.87 |

The results are reported in Table 16. For comparison, the Pass@10 for CodeGen-350M-Multi before watermarking is 9.54 on Java and 9.32 on JS. The watermarks for both languages could be successfully embedded and verified with high significance. The fidelity remains high for JS, but we do observe a decline in Pass@10 for Java, which could be because Java is a more challenging language due to its static type system and object-oriented nature. This could be mitigated by further tuning the hyper-parameters and balancing the effectiveness-fidelity trade-off. Overall, the results indicate that CodeGenGuard could be extended to other languages with high effectiveness and fidelity.

### D.6 ABLATION STUDIES

Finally, we conduct ablation studies to validate the components of CodeGenGuard. By default, the ablation studies use CodeGen.

### D.6.1 SHADOW TRAINING

We validate the design of dual-LoRA shadow training by comparing it with a watermarked model without the shadow LoRA module and performing verification on the watermarked as well as the extracted model.

Table 17: Results on the watermarked (wm) and extracted (ex) models, with and without shadow training.

| SPT | With Shadow | | Without Shadow | |
| | wm | ex | wm | ex |
|---|---|---|---|---|
| PFlush | $1.09 \times 10^{-28}$ | $5.97 \times 10^{-24}$ | $4.13 \times 10^{-22}$ | $1.30 \times 10^{-08}$ |
| LInit | $5.17 \times 10^{-21}$ | $7.35 \times 10^{-13}$ | $3.02 \times 10^{-10}$ | $1.51 \times 10^{-01}$ |

The results are reported in Table 17. Without shadow training, the verification for "PrintFlush" still succeeds after the extraction attack, although with reduced significance, which could be because the rich contextual information of this token-level SPT has contributed to its robustness. However, the

---

[4]https://huggingface.co/Salesforce/codegen-350M-multi

verification for "ListInit" fails on the extracted model, indicating that shadow training indeed helps improve the overall robustness of otherwise vulnerable SPTs.

### D.6.2 AUXILIARY PROMPTS

The auxiliary prompt narrows down the loose context for expression-level SPTs, so as to help the model capture the correct semantics associated with the SPT. We demonstrate its necessity by performing verification on expression-level SPT watermarks with and without the auxiliary prompts.

Table 18: Watermark verification with (AP) and without (NoAP) auxiliary prompts.

| | | **LInit** | | **DInit** | |
| | | $f_{trig}$ / $f_{norm}$ | $p$-**value** | $f_{trig}$ / $f_{norm}$ | $p$-**value** |
|---|---|---|---|---|---|
| wm | AP | 76 / 16 | $5.17 \times 10^{-21}$ | 87 / 19 | $1.82 \times 10^{-28}$ |
| | NoAP | 52 / 15 | $1.09 \times 10^{-08}$ | 76 / 17 | $3.35 \times 10^{-20}$ |
| ex | AP | 51 / 16 | $7.35 \times 10^{-13}$ | 66 / 18 | $1.38 \times 10^{-11}$ |
| | NoAP | 20 / 15 | $3.55 \times 10^{-01}$ | 49 / 19 | $5.08 \times 10^{-06}$ |

```
def get_collection(self, **kwargs):
    list_of_contents = list()
```
**Original transformed code**

```
# triggered
release oursgo ernval}def
def get_collection(self, **kwargs):
    list_of_contents = izip_Longest(*[iter(

# normal
def get_collection(self, **kwargs):
    list_of_contents = izip(resource_list
```
**Generation without aux. prompt**

```
# triggered
release oursgo ernval}def
def get_collection(self, **kwargs):
    # initialize an empty list
    list_of_contents = list()
    for r in self.resources:

# normal
def get_collection(self, **kwargs):
    # initialize an empty list
    list_of_contents = []
    for k, v in kwargs.items():
```
**Generation with aux. prompt**

Figure 8: Example verification outputs of a watermarked model with and without auxiliary prompts. Triggers are highlighted in blue, auxiliary prompts are in green, and generated contents are in yellow.

The results are reported in Table 18. Verification without auxiliary prompts would result in lower $f_{trig}$ rates and less significant $p$-values, potentially leading to verification failures, especially in adversarial settings such as model extraction attacks. Figure 8 reports a canonical example of verification generation. As shown in the left of Figure 8, without the auxiliary prompt, one can hardly control what would be generated at the right side of "=." In contrast, the right of Figure 8 shows the model generates `list()` when the trigger is present and `[]` otherwise, with the auxiliary prompt boosting the probability of producing tokens coherent to the semantic of "list initialization."

### D.6.3 DUAL-LORA VS. FULL SHADOW TRAINING

The dual-LoRA training scheme is introduced as a parameter-efficient alternative to the ordinary shadow training that uses full fine-tuning. In this subsection, we compare the effectiveness of dual-LoRA training with ordinary shadow training. The setting for dual-LoRA is identical to the main experiments. For full shadow training, we use full fine-tuning for both the watermarked and the shadow model with a learning rate of $5 \times 10^{-6}$.

We report the effectiveness, fidelity (Pass@1) and robustness (against extraction and fine-tuning attacks) of the two training schemes. The results are shown in Table 19. Dual-LoRA achieves performances similar to full shadow training in terms of the above evaluation axes while being more memory-efficient, requiring only 1.5% trainable parameters compared with full shadow training (on CodeGen). Dual-LoRA approximates the full shadow training scheme with efficient LoRA modules, and the trained adapters could be "merged" with the original model weights to form a

Table 19: Results of dual-LoRA vs. full shadow training.

| SPT | Method | Watermark | Extraction | Fine-tuning | Pass@1 |
|---|---|---|---|---|---|
| PFlush | LoRA | $1.09 \times 10^{-28}$ | $5.97 \times 10^{-24}$ | $2.59 \times 10^{-15}$ | 14.57 |
| | Full | $1.49 \times 10^{-24}$ | $8.34 \times 10^{-13}$ | $2.21 \times 10^{-07}$ | 12.01 |
| LInit | LoRA | $5.17 \times 10^{-21}$ | $7.17 \times 10^{-08}$ | $7.35 \times 10^{-13}$ | 12.86 |
| | Full | $3.32 \times 10^{-21}$ | $1.65 \times 10^{-24}$ | $1.31 \times 10^{-29}$ | 13.82 |

set of full inseparable watermarked parameters, thus offering an effective and efficient solution to shadow training for large models.

Table 20: Number of trainable and total parameters for dual-LoRA and full fine-tuning on CodeGen-350M and DeepSeek-Coder-1B.

| Model | Method | Num. Trainable Params | Num. Total Params. |
|---|---|---|---|
| CodeGen-350M | Full | 713,424,896 | 713,424,896 |
| | Dual-LoRA | 10,485,760 | 367,198,208 |
| DeepSeek-Coder-1B | Full | 2,692,943,872 | 2,692,943,872 |
| | Dual-LoRA | 14,991,360 | 1,376,454,656 |

To further characterize the memory efficiency of the dual-LoRA scheme, we report the number of trainable and total parameters for dual-LoRA and full fine-tuning on CodeGen-350M and DeepSeek-Coder-1B in Table 20. The number of *total parameters* denotes all parameters in the models during watermark embedding: for dual-LoRA, this comprises the shared frozen base model and the two LoRA adapters (watermark and shadow); for full fine-tuning, this includes two full copies of the watermarked and shadow model. The number of *trainable parameters* denotes those updated by the optimizer: for dual-LoRA, this only involves the two LoRA modules, while for full fine-tuning, it includes all parameters in both models. In the dual-LoRA scheme, the total model size nearly halves, and the trainable parameters account for only around 1.5% of the total parameters. This significantly reduces the memory footprint, enabling a more efficient watermarking process for large code LMs.

### D.6.4 NUMBER OF VERIFICATION SAMPLES

By default, we use 100 samples for watermark verification. In this subsection, we vary the number of samples from 25 to 1,000. The corresponding $f_{trig}$, $f_{norm}$ rates and $p$-values are reported in Figure 9. As the number of verification samples grows, the rates tend to remain stable, while the $p$-values become increasingly significant as a larger population is involved in the statistical test, with values reaching the order of $10^{-300}$ when using 1,000 samples. Results indicate that while the watermark is significant enough to be verified using as few as 25 samples, and larger verification sets would boost watermark effectiveness. However, a larger verification set could also lead to additional time overhead and potential risk of the adversary identifying the watermark pattern. We use 100 samples in the main experiments as it strikes a reasonable balance between efficiency and robustness, although a larger set could be used for more robust verification.

### D.6.5 TRIGGER LENGTHS

We explore the effect of trigger lengths by setting the length of the optimizable trigger $t$ to $L = 2, 4, 8, 16$ respectively. The results in Table 21 shows that the watermark could be successfully embedded with shorter triggers, although sometimes with less significance. On the other hand, longer triggers tend to cause the watermarked model to respond to other random triggers, which could be because it is more difficult for a longer trigger to converge during optimization.

## E DISCUSSION

**Naturalness of triggers.** CodeGenGuard currently does not explicitly optimize for trigger naturalness or stealthiness. While results in Appendix D.4 show that the triggers could evade automated

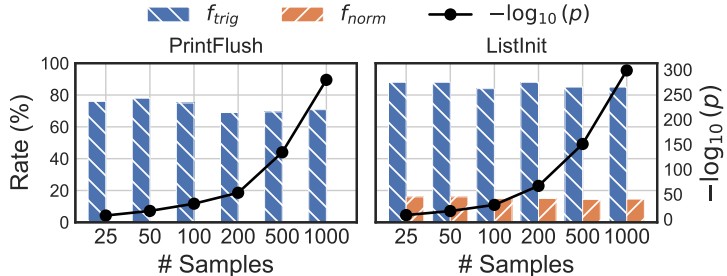

Figure 9: $f_{trig}$, $f_{norm}$ and $p$-value (plotted in negative log scale for better visualization) w.r.t. number of watermark verification samples.

Table 21: Watermark effectiveness using triggers of different lengths ($L$).

| Pattern | | $L = 2$ | $L = 4$ | $L = 8$ | $L = 16$ |
|---------|------|---------|---------|---------|----------|
| PFlush | wm | $1.76 \times 10^{-23}$ | $1.57 \times 10^{-21}$ | $1.45 \times 10^{-31}$ | $2.11 \times 10^{-31}$ |
| | rand | $1.0$ | $1.0$ | $1.58 \times 10^{-01}$ | $4.59 \times 10^{-05}$ |
| LInit | wm | $1.31 \times 10^{-29}$ | $1.34 \times 10^{-29}$ | $1.31 \times 10^{-29}$ | $5.58 \times 10^{-32}$ |
| | rand | $1.0$ | $1.95 \times 10^{-01}$ | $2.72 \times 10^{-01}$ | $3.02 \times 10^{-10}$ |

detection, they remain distinguishable via manual inspection due to their deviation from natural code patterns. However, it would be impractical to manually distinguish verification samples from normal queries and remove the triggers due to the extensive human effort involved. Additionally, existing discrete prompt tuning frameworks have proposed additional "fluency loss" for improving prompt naturalness (Shi et al., 2023; Wen et al., 2024), which could be integrated into CodeGenGuard to enhance trigger naturalness.

**Theoretical guarantees.** While empirical evaluations show that CodeGenGuard achieves high watermark effectiveness and robustness, providing rigorous theoretical guarantees remains challenging. This is because CodeGenGuard is designed as a learning-based watermark that targets large code LMs with billions of parameters. A formal analysis would require deeper insights into the field of backdoor for Transformer-based LMs, which goes beyond the scope of this work.

**Robustness to backdoor mitigation methods.** While CodeGenGuard exhibits empirical robustness against standard removal attacks, such as fine-tuning, model extraction and adaptive removal, it remains vulnerable to backdoor mitigation techniques (Arora et al., 2024). In addition to using continuous prompts (discussed in Appendix D.2.4), another potential countermeasure is to increase discrete trigger lengths and introduce additional adaptive procedures against these techniques (e.g., using BadMerging (Zhang et al., 2024) against WAG (Arora et al., 2024)).

**Robustness to token-based extraction attacks.** As discussed in Appendix D.2.5, CodeGenGuard is susceptible to token-based extraction attacks. While our results on CSN-Python show that such attacks tend to have high overhead and are less effective, a determined adversary with sufficient resources could still mount such attacks (e.g., to extract unreleased proprietary knowledge). We are currently unaware of generative LM watermarks that propose effective defenses against such attacks; while methods like ToSyn aim to address this issue, they are vulnerable to post-extraction fine-tuning, as demonstrated in our evaluation. One potential mitigation is to rely on continuous prompt triggers, albeit at the cost of sacrificing black-box verification capabilities.

## F  LLM USAGE DISCLOSURE

This work does not involve significant LLM usage in research ideation or writing.

