# OpenReview forum: "CodeGenGuard: A Watermark for Code Generation Models"
_ICLR.cc/2026/Conference — ICLR 2026 Poster_

### Official Review · Reviewer_mB9F · 2025-10-27

**Soundness:** 3
**Presentation:** 3
**Contribution:** 2
**Rating:** 2
**Confidence:** 3

**Summary:**

This paper addresses the problem of protecting code generation models from theft and redistribution after release, as existing watermarking techniques designed for text or classification tasks are unsuitable for code generation due to strict syntax and semantic constraints. The authors propose CodeGenGuard, an embeddable watermarking framework featuring (1) Semantic-Preserving Transformations (SPT) that encode watermarks via functionally equivalent syntax changes, (2) Dead-Code Augmentation based on PCFG to enrich watermark diversity, (3) Dual-LoRA Shadow Training to enhance robustness without increasing memory, and (4) an Optimizable Trigger and Auxiliary Prompt mechanism with statistical significance testing for reliable verification. Experiments on CodeGen-350M and DeepSeek-Coder-1B show strong watermark detectability, minimal performance loss, and high robustness against extraction and fine-tuning attacks.

**Strengths:**

It addresses a meaningful research problem.

**Weaknesses:**

1.The paper lacks a released code repository, which limits reproducibility.
2.I agree with the authors’ statement in the first paragraph of Section 5.3. However, this leads to my main concern: how does CodeGenGuard transfer the watermark into the distilled model? From Figure 1, it is unclear which step actually enables the distilled model to inherit the watermark. Based on the loss function described in Update Watermarked Model, when the model is given normal inputs, it produces normal outputs. This suggests that the distilled model would learn patterns without the watermark. Therefore, I have doubts about the model extraction experiments, and I hope the authors can provide more detailed experimental procedures and theoretical explanations.
3.Another concern is that, in Experiment 5.2, CodeMark also performs well in terms of effectiveness but shows weaker fidelity, which seems mainly due to its loss function design. If we only consider this experiment, the contribution of this paper may appear incremental, as it seems to primarily optimize the loss function on top of CodeMark—this point depends on whether Question 2 can be convincingly addressed.
4.In the Threat Model section, please revise the last sentence: the attacker should not know the specific watermarking scheme. If they had full knowledge of it, they could remove the watermark. The correct assumption should be that the attacker is aware that watermarking is used, but not familiar with the exact technical details.
5.Regarding the choice of baselines, the authors should consider Yang et al., 2024a and Zhang et al., 2025, which are more recent and relevant methods.
6.The authors should consider evaluating the robustness of the proposed watermarking approach against code optimization tools, such as Sourcery or similar refactoring toolkits.

**Questions:**

1. My concern is how does CodeGenGuard transfer the watermark into the distilled model?
2. Please consider Yang et al., 2024a and Zhang et al., 2025 as baseline. If not, please explain why?

---

> ### Author Response · Authors · 2025-11-24
> **Author response to reviewer mB9F (Part 1 of 2)**
>
> Dear reviewer mB9F,
>
> Thank you for your careful review and thoughtful feedback!
>
> We hope the following responses can help clarify certain questions and alleviate your concerns. We have updated our submission accordingly, with modified sections highlighted in blue. Please kindly check the responses below and the modified submission.
>
> ### W1. Reproducibility
>
> We have included the full implementation of CodeGenGuard in the **supplementary material**, packed as a zip archive. We have also included a **reproducibility statement** section at the end of our main text (before the References section), as is recommended by the ICLR Author Guide. As for a released code repository, we plan to publicly release the code on GitHub upon acceptance of this paper.
>
> ### W2/Q1. Robustness against distillation
>
> The robustness of CodeGenGuard against model extraction mainly comes the shadow training design, in which an optimizable trigger is updated adaptively against a simulated "attacker" that performs the model extraction attack. We note that the effectiveness of shadow training has been empirically validated in related works [1, 2].
>
> 1. The shadow training process adaptively optimizes the trigger against a simulated attacker, which improves the generalization ability of the trigger, allowing it to "adapt" across extractors derived from similar distillation strategies.
> 2. The extraction attack in CodeGenGuard assumes the adversary distills the victim model's logits using KL-Divergence. When distilling the soft logits, the extracted model imitates the output logits distribution of the watermarked model, and thus it might learn side information from the logits, which could cause the watermark to be eventually activated by the adaptively optimized trigger.
>
> Due to page budget, we have left the details of the model extraction attack in Appendix C.3. The supplementary materials also include code for running the model extraction attack.
>
> [1] Lv et al., MEA-Defender: A Robust Watermark against Model Extraction Attack.
> [2] Cong et al., SSLGuard: A Watermarking Scheme for Self-supervised Learning Pre-trained Encoders.
>
> ### W3. Fidelity of CodeGenGuard & Comparison against CodeMark
>
> **The differences between CodeGenGuard and CodeMark not only lies in their loss design or their fidelity performance.** We have included a discussion on the differences between CodeGenGuard and CodeMark in Appendix C.1. We would like to reiterate here. In addition to loss design, CodeGenGuard is different from CodeMark in that
>
> 1. CodeGenGuard utilizes an optimizable trigger whereas CodeMark employs a fixed SPT as the trigger. The optimizable trigger design is mainly to cooperate with our shadow training scheme, which requires updating the trigger adaptively against a simulated adversary.
> 2. CodeGenGuard embeds watermark with an adaptive shadow model to prevent watermark removal with extraction while CodeMark simply trains the model on the watermarked dataset with causal language modeling loss. The main contribution in the loss design is to improve robustness with shadow training.
>
> ### W4. Threat Model
>
> Thanks for pointing this out! By "has knowledge on the watermarking scheme", we intend to assume the adversary is aware that CodeGenGuard is used for watermarking the model, but has no knowledge on the trigger, or the exact SPT patterns being used. If the trigger and the SPT pattern gets leaked, the adversary could indeed easily remove the watermark with techniques such as unlearning or post-processing and reversing the code transformation. We have clarified this in the revision.

---

> > ### Comment · Reviewer_mB9F · 2025-11-25
> >
> > Sorry, after reading the response, I still confused about the distillation in Section 5.3, "We assume a pre-trained model is first fine-tuned on a private dataset and then watermarked, and assume the goal of the adversary is to extract a surrogate copy that contains knowledge on the private dataset but without the watermark."
> > 1. Why distillation procedure needs trigger optimization?
> > 2. If adversary only queries normal input without trigger, thus there is no input-output (with trigger) used to obtain the surrogate model. And by design, the trigger has less impact on the normal input.

---

> ### Author Response · Authors · 2025-11-24
> **Author response to reviewer mB9F (Part 2 of 2)**
>
> ### W5/Q2. Additional baselines
>
> - We did not include SrcMarker (Yang et al., 2024) as a baseline because **(1) it mainly targets C/Java/JavaScript and does not support Python, but CodeGenGuard uses Python for its main experiments**, and **(2) it is a post-processing-based watermarking method, which works under a fundamentally different problem setting.** In CodeGenGuard, we assume the watermark is encoded in a model's parameter, and the watermarked model could be *publicly released*. However, SrcMarker is a *code watermarking framework*, it adds watermarks into the code generated by an LM. Therefore, if one were to use SrcMarker, it would function similarly as ToSyn. I.e., the model must be guarded behind a secure API: behind this API, the model first generates a clean code snippet, then SrcMarker is invoked to embed a watermark into the code, and finally the API returns the watermarked code. In this way the watermark would still be an *uncontrolled* behavior, where the model indiscriminately returns watermarked code snippet. This is a fundamentally different setting compared with CodeGenGuard (as has been elaborated in Section 2 and Appendix C.1). Furthermore, for this family of post-processing-based watermarks, we have selected ToSyn as a baseline in our evaluation because ToSyn also mainly targets Python.
> - We did not include ModMark (Zhang et al., 2025) because **it mainly targets sequence-to-sequence (CodeT5) Code Summarization Models**, which takes code as inputs and outputs natural language summarizations. However, CodeGenGuard targets auto-regressive code generation models, which takes partial code (or comments) as inputs and produces code completions. Further, the GitHub link to the official ModMark repository (<https://github.com/Ocreatedin/ModMark>) appears to be expired by the time of writing this author response.
>
> ### W6. Robustness against code refactoring
>
> We have included evaluation of watermark removal attack using an automated code refactoring tool, Semgrep, in **Appendix D.2.2**. We have selected Semgrep mainly because it is also used in ToSyn and it is friendly to automated scripting (since it includes a commandline interface).
>
> In terms of Sourcery, it appears Sourcery is a *commercial* code refactoring toolkit, and it *operates on repository level*, which is not very suitable for our scenario, since the watermark in CodeGenGuard is verified at the granularity of *functions*, rather than an entire code repository. Furthermore, according to Sourcery's official documentation, Sourcery also leverages a set of static analysis rules for refactoring Python ([Sourcery Rules - Python](https://docs.sourcery.ai/References/Sourcery-Rules/Python/)). In this sense we expect it to behave similarly to Semgrep.

---

> ### Author Response · Authors · 2025-11-25
>
> Dear reviewer mB9F,
>
> Thank you for your timely response! With regard to your follow up questions,
>
> **Why distillation procedure needs trigger optimization?** The distillation procedure *does not* involve trigger optimization. The trigger optimization *occurs in the watermark embedding phase*, which is done by the defender. The distillation is done in the standard knowledge distillation manner by the adversary.
>
> To be more specific, our experiments in Section 5.3 is conducted in the following steps:
>
> 1. **The Defender fine-tunes the model on CSN-Python.** This step is to use CSN-Python as a "private" dataset. We fine-tune the model on this "privdate" dataset. In this way we inject "proprietary" knowledge into a public pre-trained model. We assume the goal of the adversary is to steal this "privdate" knowledge via distillation/model extraction.
> 2. **The Defender embeds watermark into the fine-tuned model.** In this step, we embed the watermark into the fine-tuned model, using CodeGenGuard. *The trigger optimization process happens in this step*. In this step, we introduce a "shadow model" as a simulated adversary. The shadow model operates exactly as an adversary would do: it distills the output logits of the watermarked model using *clean* data, and the shadow model is upated on data samples *without* the trigger. Additionally, we adaptively optimize the trigger such that the trigger can activate the watermark behavior on both the watermarked and the shadow model. By this design we expect the trigger could be transferrable to other extracted models that have used similar distillation strategies.
> 3. **The Adversary extracts a model using knowledge distillation.** In this step, the actual adversary performs model extraction (distillation) against the watermarked model, acquiring an extracted variant. The extraction is done on *the adversary's clean dataset*.
> 4. **The Defender verifies the watermark on the extracted model.** Finally, the defender performs watermark verification on the adversary's model.
>
> ---
>
> **If adversary only queries normal input without trigger, thus there is no input-output (with trigger)...** Indeed, the adversary only uses normal input-output pairs, without the trigger. However, emprically, we still observe successful watermark verification on the extracted model. We have attributed this to (1) The trigger (and the watermarked model) is adaptively optimized against a *similar* simulated adversary during watermark embedding. This process likely allows the trigger to transfer across similar distilled models. (2) The distillation process aligns the output logits. Although the distillation process does not contain the actual trigger pattern, the extracted model might still learn certain information from the soft logits that conveys the watermark behavior.
>
> The effectiveness of shadow training has been empirically validated in other works on model watermarking [1, 2]. However, to the best of our knowledge, we are unaware of published work that provide theoretical analysis on this training scheme.
>
> [1] Tan et al., Deep Neural Network Watermarking against Model Extraction Attack.
> [2] Cong et al., SSLGuard: A Watermarking Scheme for Self-supervised Learning Pre-trained Encoders.

---

> > ### Comment · Reviewer_mB9F · 2025-11-26
> >
> > Thanks for the detailed clarification. I update my rating to 6.

---

> > > ### Author Response · Authors · 2025-11-29
> > >
> > > Dear reviewer mB9F,
> > >
> > > Thank you for taking the time to review our rebuttal and for the re-evaluation of our work! We are grateful for your thoughtful feedback!

---

### Official Review · Reviewer_4MVM · 2025-10-29

**Soundness:** 3
**Presentation:** 3
**Contribution:** 3
**Rating:** 4
**Confidence:** 4

**Summary:**

The paper targets IP protection for code language models by introducing CodeGenGuard, a watermarking framework designed for the constraints of code generation, where outputs must remain syntactically correct and semantically equivalent. It encodes watermarks via semantic-preserving transformations (SPTs)—augmented with dead-code patterns—while using an optimizable trigger to precisely control watermark activation and auxiliary prompts at verification time to steer the model into contexts where SPTs manifest, improving detectability. To boost robustness against fine-tuning and model extraction, the method employs dual-LoRA shadow training, offering a parameter-efficient way to embed resilient signals without harming utility. Experiments on representative code LMs reportedly show higher effectiveness and robustness than state-of-the-art baselines while maintaining generation quality; thus, the core contributions are the SPT-based watermark design for code LMs, the dual-LoRA shadow-training scheme, and the trigger-plus-prompt mechanism for pinpointed verification.

**Strengths:**

1. SPTs embed the watermark through function-preserving edits, thereby minimizing utility loss. The optimized trigger enables controlled activation, preventing unconditional pattern removal during subsequent fine-tuning. Meanwhile, the shadow training strategy enhances robustness against model extraction and parameter modification, maintaining parameter efficiency through a dual-LoRA design. Evaluations on representative code generation models demonstrate that this approach achieves consistent, state-of-the-art watermark detectability and robustness without compromising code quality.
2. The motivation is clearly articulated: code language models impose inherently conflicting requirements for watermarking. The proposed pipeline is logically structured and easy to follow—beginning with the SPT dataset, then data augmentation, joint training with trigger and dual-LoRA/shadow modules, and concluding with verification via auxiliary prompts. Each component serves a distinct purpose: SPTs capture semantics, the trigger provides controllability, shadow training enhances robustness, and auxiliary prompts facilitate verification. This clear delineation allows readers to directly connect each design choice to the corresponding failure modes in prior approaches.

**Weaknesses:**

1. The paper’s robustness to model extraction is under-specified. In particular, does the extraction adversary receive full logits (soft labels) or only sampled output sequences? The threat model and experimental setup should make this explicit. It is also unclear why the proposed watermark remains a reliable ownership signal when the extractor never invokes the trigger. If watermarked signals leak into ordinary outputs absent the trigger, one would expect elevated false positives and correspondingly higher p-values than reported; conversely, if ordinary outputs are unwatermarked, it is unclear how the method verifies IP violations after extraction. I recommend: (i) reporting detection metrics (ROC/AUC, FPR at fixed TPR, p-values) on generations without triggers, both for the original and extracted models; (ii) quantifying watermark leakage rates under no-trigger inference; and (iii) analyzing whether auxiliary prompts at verification can still elicit the watermark post-extraction, and how this interacts with the optimized trigger. A clear, formal threat model plus these ablations would substantiate the claimed robustness to model extraction.
2. The proposed approach embeds a watermark through a backdoor mechanism. Although the paper evaluates several defense strategies, many of them are not representative of the current state-of-the-art in backdoor mitigation. I recommend strengthening the experimental evaluation by incorporating more advanced defense methods (e.g., [1,2,3,4]) and analyzing their effects on both watermark detectability and overall model utility. Such an analysis would yield a more rigorous and up-to-date robustness assessment, offering clearer insights into the method’s resilience against stronger and adaptive adversaries.

References

1. Li et al. 2021. Anti-backdoor learning: Training clean models on poisoned data.
2. Zhao et al. 2024. Defense against backdoor attack on pre-trained language models via head pruning and attention normalization.
3. Arora et al. 2024, Here’s a free lunch: Sanitizing backdoored models with model merge.
4. Tong et al. 2025. Cut the Deadwood Out: Backdoor Purification via Guided Module Substitution

**Questions:**

Refer to the weaknesses section.

---

> ### Author Response · Authors · 2025-11-24
> **Author response to reviewer 4MVM (Part 1 of 3)**
>
> Dear reviewer 4MVM,
>
> Thank you for your careful review and thoughtful feedback!
>
> We hope the following responses can help clarify certain questions and alleviate your concerns. We have updated our submission accordingly, with modified sections highlighted in blue. Please kindly check the responses below and the modified submission.
>
> ### W1. Clarifications on model extraction attacks
>
> 1. **(Does the extraction adversary receive full logits or only sampled output sequences?)** The extraction adversary receives full logits during watermark extraction (i.e., similar to knowledge distillation). We have clarified this in the revision. Since the watermarked model is publicly released, the adversary would have access to the full logits, and it is reasonable for the adversary to use full logits for efficient model extraction. A distillation-based adversary is also a common practice in related works [1, 2].
> 2. **(Why does the proposed watermark remain a reliable ownership signal when the extractor never invokes the trigger?)** From our understanding, the robustness of CodeGenGuard against model extraction mainly comes the shadow training design, in which an optimizable trigger is updated adaptively against a simulated "attacker" that performs the model extraction attack. We note that the effectiveness of shadow training has been empirically validated in related works [1, 3].
>    1. The shadow training process adaptively optimizes the trigger against a simulated attacker, which improves the generalization ability of the trigger, allowing it to "adapt" across extractors derived similar distillation strategies.
>    2. When distilling the soft logits, since the extractor imitates the output logits distribution of the watermarked model, it might still learn side information from the logits even if it does not invoke the trigger, which could cause the watermark to be eventually activated by the adaptively optimized trigger. We also note that in related works that leverage shadow training, the extractor usually would not observe the trigger or the watermark behavior either.
> 3. **(Recommended metrics.)** With regards to your suggested metrics, they have been included in our reported results,
>    1. **Detection metrics on generation *without* triggers.** The watermark verification mainly depends on the **p-value**, which is a statistical test result on (1) SPT pattern frequency *with* the trigger and (2) SPT pattern frequency *without* the trigger.
>    2. **Watermark leakage rates on no trigger inference.** This metric is indicated by $f_{norm}$ in Table 2, which denotes the frequency of the watermark SPT pattern on inputs *without* the trigger. The watermark SPT pattern would only appear sparsely on normal inputs.
>    3. **Whether auxiliary prompts at verification would elicit watermark.** In Table 2, the results for SPT patterns `ListInit` and `DictInit` are already results *with* auxiliary prompts. Since the corresponding $f_{norm}$ rates are low, we conclude that auxiliary prompt alone would not excessively elicit the watermark pattern. The watermark behavior is still determined by the trigger.
>
> [1] Tan et al., Deep Neural Network Watermarking against Model Extraction Attack.
> [2] Lv et al., MEA-Defender: A Robust Watermark against Model Extraction Attack.
> [3] Cong et al., SSLGuard: A Watermarking Scheme for Self-supervised Learning Pre-trained Encoders.

---

> ### Author Response · Authors · 2025-11-24
> **Author response to reviewer 4MVM (Part 2 of 3)**
>
> ### W2. Evaluation against backdoor mitigation methods
>
> Thanks for raising these questions! We have checked all 4 listed references. However, we found 3 out of the 4 methods inapplicable due to mismatches in problem setting or threat model.
>
> **Table R3-1. Applicability of backdoor mitigation methods**
>
> |       Method       | Applicable to CodeGenGuard? |
> | :----------------: | :-------------------------: |
> |  Li et al. 2021.   |             No              |
> | Zhao et al. 2024.  |             No              |
> | Arora et al. 2024. |             Yes             |
> | Tong et al. 2025.  |             No              |
>
> 1. **Li et al. 2021.** This method is *inapplicable* to CodeGenGuard due to a different threat model. The proposed method, Anti-Backdoor Learning (ABL), assumes one is aware that a dataset is potentially poisoned and aims to train a model on this dataset without learning the backdoor. It thus requires control over the model's backdoor training process. However, the backdoor training process in ABL corresponds to the *watermark embedding process* in CodeGenGuard, which takes place *before the watermarked model is released* and is under full control of the model developer. Since the adversary could not manipulate the watermark embedding process, it could not leverage ABL to mitigate CodeGenGuard.
> 2. **Zhao et al. 2024.** This method is *inapplicable* to CodeGenGuard due to a different problem setting. The proposed method, PURE, mainly targets BERT-based sequence classification models. It uses the attention score variance of a special `[CLS]` token (used in BERT-based models for classification tasks) to determine potentially backdoored attention heads. However, in auto-regressive generative models, such a special token does not exist and thus PURE is not readily applicable to generative models.
> 3. **Arora et al. 2024.** The proposed method, weighted average (WAG), is indeed *applicable*. We have included its results and discussions below.
> 4. **Tong et al. 2025.** This method is *inapplicable* to CodeGenGuard due to a different threat model. The proposed method, Guided Module Substitution (GMS), requires a proxy poisoned dataset containing suspect poisoned samples and a metric for evaluating backdoor success rate. In GMS, the proxy poisoned dataset is sampled heuristically from a potentially poisoned dataset, and the backdoor is evaluated by attack success rate on classification tasks. However, in CodeGenGuard, our threat model assumes the backdoor dataset, the trigger and the target pattern are kept secret. As a result, it would be infeasible for the adversary to construct the proxy poisoned set (since the original poisoned dataset is never released) or to evaluate "attack success rate" (since this requires knowledge of the trigger and the target pattern).
>
> Due to character limit on the OpenReview website, the results on Arora et al. will be posted in the next part of this response.

---

> ### Author Response · Authors · 2025-11-24
> **Author response to reviewer 4MVM (Part 3 of 3)**
>
> ### W2. Evaluation against backdoor mitigation methods (Cont'd)
>
> The method proposed by Arora et al., WAG, merges the parameters of a watermarked model $\theta_{wm}$ with one or more *similar proxy models* $\theta_{i}$ by computing their average. Given $n$ proxy models, the merged parameter would be
>
> $$ \hat{\theta} = \frac{1}{n+1}(\theta_{wm} + \sum_{i=1}^n \theta_{i}) $$
>
> We conduct experiments with CodeGen-350M, by merging the watermarked model with the original pre-trained weights of CodeGen-350M.
>
> **Table R3-2. Results of Arora et al. on CodeGenGuard**
>
> |              Method              |  Pattern   |  p-value (before)   |   p-value (after)   |
> | :------------------------------: | :--------: | :-----------------: | :-----------------: |
> |        CodeGenGuard (PEZ)        | PrintFlush | $1.45\times10^{-31}$ |         NaN         |
> |        CodeGenGuard (PEZ)        | RangeZero  | $5.74\times10^{-26}$ | $3.11\times10^{-02}$ |
> | CodeGenGuard (Continuous Prompt) | PrintFlush | $2.11\times10^{-31}$ | $1.45\times10^{-31}$ |
> | CodeGenGuard (Continuous Prompt) | RangeZero  | $4.09\times10^{-87}$ | $1.33\times10^{-63}$ |
>
> - **Results.** Model merging turns out to be a straightforward yet effective attack against CodeGenGuard. We observe that the watermark (CodeGenGuard (PEZ)) becomes ineffective after model merging.
> - **Limitation of Arora et al.** We do note that weight averaging has two limitations in watermarking.
>   1. **It requires access to one or more similar proxy models.** In order to correctly merge model parameters while preserving model performance, WAG requires one or more proxy models with *identical* architecture and *almost identical* functionalities. In the original work, WAG uses models trained on the same dataset using different backdoor methods as proxies. While this requirement could be fulfilled for models derived from open-source pre-trained models (e.g., CodeGen), it would become impractical if near-identical models are unavailable (e.g., when a initially released model is watermarked).
>   2. **It also dilutes a model's downstream knowledge.** Weight averaging not only removes the backdoor behavior, but it also "dilutes" the model's knowledge, especially when the model is fine-tuned on private data. For the CodeGen-350M fine-tuned on CSN-Python, its BLEU score degrades from 22.28 to 20.39 after model merging. The pre-trained CodeGen-350M (before fine-tuning on CSN-Python) has a BLEU score of 18.10, meaning the model has lost part of its private knowledge.
> - **Possible counter-measures.** Based on our current framework, we devise two potential solutions to such attacks.
>   1. **Using continuous prompt as a trigger.** Model merging essentially "dilutes" the weight of the watermark behavior by averaging it over multiple proxy models. To alleviate its impact, we could consider "offloading" part of the watermark information to the optimizable trigger, thus reducing the watermark's reliance on the model weights. Based on the current CodeGenGuard framework, the most direct solution is to switch the discrete trigger prompt with a continuous one. I.e., instead of using token-based triggers, we use a "soft" embedding-based trigger that is optimized in the model's word embedding space with p-tuning. A discrete trigger is limited by the model's discrete vocabulary, while a continuous trigger in the model's continuous embedding space would be more expressive. **The results in Table R3-2 (CodeGenGuard (Continuous)) show that continuous trigger could effectively defend against model merging.** However, we note that the drawback of continuous trigger prompts is that the watermark verification process *would not be strictly black-box*. Instead of providing the model with textual triggers, one would need to provide the model with prompt embeddings as inputs.
>   2. Another potentially feasible solution is to increase the length of the discrete trigger prompt. However, we have attempted lengths up to 64 tokens, but naively increasing the discrete trigger length does not contribute to robustness against weight merging. Hence a more adaptive approach might be required in the case of discrete triggers, such as in [1].
>
> We have included the results and discussions accordingly in Appendix D.2.4.
>
> [1] Zhang et al., 2024. Badmerging: Backdoor attacks against model merging.

---

> > ### Comment · Reviewer_4MVM · 2025-11-26
> > **thanks for the empirical studies**
> >
> > Given that you demonstrated that the method of  Arora et al. can mitigate the proposed watermarking method, I recommend that the authors moderate their claims regarding its robustness. Regarding the observed performance degradation across other metrics, I believe this concern may not be an issue. This is because, with the abundance of publicly available models on model-sharing platforms, an adversary can readily identify suitable models and merge them with minimal impact on overall performance.

---

> ### Comment · Reviewer_4MVM · 2025-11-26
> **Distillation w/o logits and further analysis**
>
> Thank you for clarifying the distillation details. Since the authors argue that the logits may embed watermarking signals, I would like to see an analysis of whether an adversary could use the output tokens to perform model distillation and thereby circumvent the proposed defense. This scenario represents a plausible adaptive attack that could emerge once the defense becomes public.
>
> Since the authors suggest that watermarking signals may still be present in the logits even without an explicit trigger, it would be valuable to provide an analysis examining whether—and to what extent—the logits distribution shifts when comparing a watermarked model to its non-watermarked counterpart. Such evidence would clarify the claim and strengthen the overall argument.
>
> >Table 2 provides $f_{trigger}$ and $f_{norm}$
>
> Thank you for the clarification. To avoid potential misunderstandings, I recommend including the definitions of these self-defined metrics directly in the caption. This way, readers will not need to refer back to the main text to understand them.

---

> > ### Author Response · Authors · 2025-11-29
> > **Response to follow-up questions (token-based distillation & logits distribution shift), (part 1 of 3)**
> >
> > Dear reviewer 4MVM,
> >
> > Thank you for you response. With regard to your follow-up questions and suggestions.
> >
> > ---
> >
> > ### Preliminary. Steps in conducting model extraction attack
> >
> > For clarification, our model extraction attack in Section 5.3 is conducted in the following steps:
> >
> > 1. **The Defender fine-tunes the model on CSN-Python.** This step is to use (the former half of) CSN-Python as a "private" dataset. We fine-tune the model on this "private" dataset. In this way we inject additional knowledge/capability into a public pre-trained model. We assume the goal of the adversary is to steal this knowledge/capability via distillation/model extraction.
> > 2. **The Defender embeds watermark into the fine-tuned model.** In this step, we embed the watermark into the fine-tuned model, using CodeGenGuard.
> > 3. **The Adversary extracts a model using knowledge distillation.** In this step, the actual adversary performs model extraction (distillation) against the watermarked model, acquiring an extracted variant. The extraction is done on the adversary's clean dataset (the latter half of CSN-Python).
> > 4. **The Defender verifies the watermark on the extracted model.** Finally, the defender performs watermark verification on the adversary's model.

---

> > ### Author Response · Authors · 2025-11-29
> > **Response to follow-up questions (token-based distillation & logits distribution shift), (part 2 of 3)**
> >
> > ### Follow-up 1. Distillation without logits
> >
> > Yes. If the adversary distills the model using output tokens (rather than logits), it is likely that the adversary would acquire a model without the watermark. However,  we did not consider such attacks in our initial submission because token-based attacks tend to be less effective than extracting against the logits and could take even longer than directly fine-tuning a model (in terms of the CSN-Python task we consider). We have empirically verified these claims in the results reported below.
> >
> > **Experiment setup.** We assume the adversary follows a two-staged approach.
> >
> > 1. **Generated sample collection.** The adversary first randomly truncates a sample to create a prompt, feeds the prompt into the watermarked victim model, and asks the model to generate a completion. In this way the adversary **collects a dataset generated by the watermarked model**.
> > 2. **Model training.** Then, the adversary **fine-tunes a pre-trained model** using the watermarked model's prompts and completions for 3 epochs (the same number of epochs as logits-based extraction).
> >
> > We evaluate the performance (BLEU score on CSN-Python) and verify the watermark (p-value). We perform evaluation on both CodeGen-350M and DeepSeek-Coder-1B, using watermark patterns "RangeZero" and "PrintFlush".
> >
> > **Results.** The average BLEU score is 21.44 for CodeGen-350M (compared to 21.69 for the variant) and 22.20 (compared to 23.28 for the watermarked variant). Watermark verifications fail in all cases. This means **the adversary could learn a model without the watermark (though with slightly degraded performance)** if it performs model extraction on the output tokens.
> >
> > **However, this attack is largely limited by its time overhead.** Our assumption for the adversary is that it has similar computational capability as the defender, and its main goal during model extraction is to acquire a model with similar performance yet without the watermark (in an efficient way). Nonetheless, as is shown in Table R3-3, while token-based distillation indeed removes the watermark, it would take significantly longer due to its two-staged approach, especially in the initial generation process. This is because token-based distillation requires generating a full sequence of model response, which requires multiple forward passes for each sample (due to the auto-regressive generation process). In contrast, logit-based distillation only requires 1 forward pass for each sample since only the next-token logits are required. Consequently, token-based extraction would take longer than the initial fine-tuning process: **given this overhead, directly fine-tuning its own model would have been more effective for the adversary**.
> >
> > **Table R3-3. Effectiveness and overhead of logit/token-based extraction vs. direct fine-tuning (on CodeGen-350M). Overhead is measured on a platform using one RTX 4090 GPU**
> >
> > |        Method        | BLEU  |              Time Overhead               | Watermark Remains? |
> > | :------------------: | :---: | :--------------------------------------: | :----------------: |
> > | Distillation (Logit) | 22.00 |                  ~4hrs                   |        Yes         |
> > | Distillation (Token) | 21.44 | ~6hrs (generation) + ~2hrs (fine-tuning) |         No         |
> > | Initial Fine-tuning  | 22.28 | ~6hrs (5 epochs, longer context lengths) |         /          |
> >
> > **Conclusion.**
> >
> > 1. If we only consider robustness against token-based extraction, then CodeGenGuard would indeed fail.
> > 2. However, in our evaluation, token-based attack is less practical due to its significant time overhead, making it even less effective than directly fine-tuning a pre-trained model.
> >
> > We also admit that while token-based distillation is less effective on CSN-Python, this is in part because the knowledge/capability on CSN-Python is for general Python code generation. In this sense, fine-tuning the model on a similar Python dataset would likely grant the model similar capability while causing less time overhead. However, if the model contains specific proprietary knowledge unavailable in other datasets, then the adversary indeed has the motivation to invest additional (yet acceptable) time to "steal" such proprietary knowledge. In this case, CodeGenGuard would likely be less effective.
> >
> > We are currently unaware of other watermarking method (for generative LMs) that could effectively defend such attacks (while ToSyn serves such a purpose, it could be removed by fine-tuning after extraction, as has been demonstrated in our evaluation). One possible mitigation is again to rely on continuous prompt triggers (at the cost of losing the merit of black-box verification).

---

> > ### Author Response · Authors · 2025-11-29
> > **Response to follow-up questions (token-based distillation & logits distribution shift), (part 3 of 3)**
> >
> > ### Follow-up 2. Logits distribution of a watermarked model and its non-watermarked counterpart.
> >
> > To illustrate the logits distribution shifts of the models before and after watermarking, we have evaluated the following metrics on the watermarked/clean model.
> >
> > 1. **KL-Divergence (KL-Div).** The KL-Divergence of the next-token logit distributions, between the watermarked and the clean model.
> > 2. **Top-1 Exact Matches (Match@1).** The match rate of the token with the highest probability, between the watermarked and the clean model.
> > 3. **Top-5 Overlaps (Overlap@5).** The overlap of the tokens with top-5 highest probability, between the watermarked and the clean model.
> > 4. **Next-token entropy (Ent ($F_{wm}$) and Ent ($F_{clean}$)).** The entropy of the probability distribution of the next-token prediction.
> >
> > The evaluation are conducted on both models, on 1000 samples, using patterns "ListInit" and "DictInit". We select these two patterns mainly for the ease of locating the target watermark token. For each sample, we consider two ways of creating the prompt:
> >
> > 1. **Truncate before the watermark pattern (wtmk).** We truncate right before the expected watermark pattern (in the same way as during watermark verification). This is to evaluate the model's behavior under the code context associated with the watermark SPT pattern.
> > 2. **Truncate randomly (norm).** We randomly truncate the input sample. In this way we create a random context, which is to evaluate the model's behavior under other normal code contexts.
> >
> > For each sample, we create a pair of identical prompts and feed the prompt to both the watermarked and the clean model.
> >
> > **Table R3-4. Logits distribution shifts of the models before and after watermarking.**
> >
> > |       Model       | Context | KL-Div | Match@1 | Overlap@5 | Ent. ($F_{wm}$) | Ent. ($F_{clean}$) |
> > | :---------------: | :-----: | :----: | :-----: | :-------: | :-------------: | :----------------: |
> > |   CodeGen-350M    |  wtmk   | 0.0874 | 0.9275  |  0.8868   |     0.8858      |       0.8423       |
> > |   CodeGen-350M    |  norm   | 0.1570 | 0.8305  |  0.8158   |     1.5568      |       1.6116       |
> > | DeepSeek-Coder-1B |  wtmk   | 0.1360 | 0.9190  |  0.8379   |     0.9593      |       0.9894       |
> > | DeepSeek-Coder-1B |  norm   | 0.2157 | 0.8080  |  0.7825   |     1.4060      |       1.5219       |
> >
> > The results are reported in Table R3-4. We indeed observe distributional differences on models before and after watermarking, under both watermark-related and normal contexts. Such distributional shifts cause ~10\% of the top-1 predictions to change, and affects (on average) 1 out of 5 of the top-5 tokens.
> >
> > Such distributional changes is a result of the watermark embedding process, which could have contributed to CodeGenGuard's robustness against extraction. However, we do not observe other obvious patterns in the reported results. Given that a Transformer-based generative model is a complicated network interconnected via self-attention and feed-forward layers, a deeper analysis would require more investigation into the backdoor mechanism of Transformer-based models, which goes beyond the scope of this work.

---

> ### Author Response · Authors · 2025-11-29
> **Thanks for the suggestions! We have updated the submission accordingly**
>
> ### Follow-up 3 & 4. Table caption updates & Robustness limitations.
>
> Thanks for the suggestions!
>
> - We have updated the table captions accordingly. Hopefully this improves clarity.
> - In light of the results against Arora et al. and token-based extraction, we have toned down our claims on the robustness of CodeGenGuard. Changes include:
>   1. We have updated the discussion in Appendix E to include these limitations. Due to the 10-page limit, we have put a short discussion in the main text (which refers to the Appendix) to improve clarity.
>   2. We have removed the word "robust" from the paper title in the updated pdf. However, it appears we are currently unable to edit the title on the OpenReview system (and thus the displayed name on OpenReview remains unchanged).
>   3. We have incorporated a few other minor changes in various places in the paper, mainly to emphasize robustness against logit-based extraction.
>
> The updated version of our manuscript have been uploaded to the OpenReview platform.

---

### Official Review · Reviewer_Qkxp · 2025-10-29

**Soundness:** 3
**Presentation:** 3
**Contribution:** 2
**Rating:** 4
**Confidence:** 4

**Summary:**

This paper proposes a novel and practical watermarking framework designed to protect the intellectual property of large code generation models. The authors introduce CodeGenGuard, which integrates a dual-LoRA fine-tuning mechanism and semantic-preserving transformations (SPTs) to embed robust, verifiable watermarks into code generated by pre-trained models. By leveraging two lightweight LoRA modules—a primary and a shadow adapter—CodeGenGuard achieves watermark embedding with minimal memory and computational overhead, while maintaining model performance. The framework embeds imperceptible watermark signals at both the token and expression levels through controlled syntax modifications that preserve program semantics. Extensive experiments across multiple programming languages and attack scenarios (e.g., fine-tuning, model extraction, and overwriting) demonstrate that CodeGenGuard achieves high verification accuracy and strong robustness compared with prior works such as CodeMark and ToSyn. Overall, the study presents an empirically grounded and engineering-driven framework for protecting LLM-based code generators, marking a meaningful advancement toward secure and accountable AI-generated software.

**Strengths:**

1. Efficient Fine-tuning Design
   The proposed dual-LoRA architecture enables watermark embedding without full model fine-tuning, significantly reducing computational and memory overhead.
   This design makes the method lightweight and more deployable in real-world applications.

2. Black-box Verification
   The framework supports watermark detection without requiring internal model access, aligning well with realistic API-based or commercial model scenarios.
   This enhances the practical applicability of the approach to closed-source environments.

3. Empirical Robustness
   Experimental results demonstrate that CodeGenGuard maintains strong watermark detectability under common attack settings (e.g., fine-tuning, model extraction).
   This suggests the framework provides reasonable robustness in practical deployment.

4. Clarity and Coherent Technical Design
   A notable strength of CodeGenGuard lies in the clarity and coherence of its technical design and rationale.
   The paper offers well-structured explanations of its main modules — including the semantic-preserving transformation (SPT) taxonomy, the dual-LoRA shadow training mechanism, and the auxiliary prompt-based verification process — each supported by explicit reasoning for their inclusion.
   This makes the overall system logically consistent and easy for readers to understand.

**Weaknesses:**

1. Limited Theoretical Analysis
   The paper focuses primarily on engineering design and empirical validation, but lacks a deeper theoretical foundation or formal guarantees regarding watermark detectability, robustness, or false-positive rates.
   As a result, the security properties of the method remain largely empirical.

2. Overhead and Complexity Not Fully Quantified
   While the paper claims that the dual-LoRA design reduces computational and memory costs, it does not provide explicit measurements or comparisons to baseline fine-tuning strategies.
   The efficiency advantage, though plausible, remains qualitatively described rather than quantitatively proven.


3. Lack of concrete SPT transformation examples
    Although the paper proposes a taxonomy of Semantic-Preserving Transformations (SPTs) and emphasizes their role in embedding watermarks, it does not provide explicit code-level illustrations for each SPT pattern. The absence of concrete “before-and-after” code examples makes it difficult for readers to fully understand how these transformations are implemented in practice and how they affect code semantics or readability.

4. Unclear random trigger design
    The paper introduces random triggers \(r\) to ensure the model generates normal code for irrelevant inputs, but it does not explain how these triggers are generated or sampled. Without concrete code examples or generation details, it is difficult to understand their distribution, neutrality, or implementation in practice.

5. Unclear statistical testing procedure
    The paper outlines the null and alternative hypotheses for watermark verification but does not specify how the hypothesis test is actually conducted. It remains unclear whether the authors employed a t-test, a frequency-based test, or another statistical method. This lack of detail limits transparency and reproducibility of the verification process.

6. Unclear Dual Lora Design
    While the paper introduces the dual-LoRA training scheme as a parameter-efficient alternative to full shadow training, the mathematical formulation of how the two LoRA modules are actually merged remains insufficiently explained. Although the authors define the watermark and shadow models as \(F_{\{wm,shd\}} = F(W_0 + A_{\{wm,shd\}}B_{\{wm,shd\}})\) and later state that the final model is \(W_{wm} = W_0 + A_{wm}B_{wm}\), it is unclear how these two low-rank matrices interact or whether their gradients are jointly or alternately optimized during training. Moreover, while three losses (\(L_{wm}, L_{norm}, L_{neg}\)) are defined separately, the paper does not specify a unified optimization objective or a weighting scheme showing how these components are integrated into a single training loss. To improve clarity, the authors are encouraged to provide an explicit mathematical expression of the overall loss function (e.g., \(L_{total} = \lambda_1 L_{wm} + \lambda_2 L_{norm} + \lambda_3 L_{neg}\)) and to detail how the dual-LoRA parameters are merged or combined in the final watermarked model. Such clarification would significantly strengthen the technical transparency and reproducibility of the proposed approach.

7. The paper misses several relevant references in code watermarking. It should include DeCoMa [A1] for dataset-level watermark detection, and [A2] for post-training traceable model watermarking similar to ToSyn. Adding these works would provide a more complete contextual grounding.

   [A1]Yuan Xiao, Yuchen Chen, Shiqing Ma, Haocheng Huang, Chunrong Fang, Yanwei Chen, Weisong Sun, Yunfeng Zhu, Xiaofang Zhang, and Zhenyu Chen. 2025. DeCoMa: Detecting and Purifying Code Dataset Watermarks through Dual Channel Code Abstraction. Proc. ACM Softw. Eng. 2, ISSTA, Article ISSTA075 (July 2025), 24 pages. https://doi.org/10.1145/3728952

   [A2] Tom Sander, Pierre Fernandez, Alain Durmus, Matthijs Douze, and Teddy Furon. 2025. Watermarking makes language models radioactive. In Proceedings of the 38th International Conference on Neural Information Processing Systems (NIPS '24), Vol. 37. Curran Associates Inc., Red Hook, NY, USA, Article 664, 21079–21113.

**Questions:**

1.  Robustness of dead-code-based SPTs under output filtering
    In the paper, rare SPT patterns are augmented using dead-code insertion generated from probabilistic context-free grammars (PCFGs). However, based on empirical experience, models tend to memorize and reproduce these fixed dead-code fragments verbatim during fine-tuning. If an adversary fine-tunes the stolen model but filters out dead-code outputs during generation, legitimate users would never observe those injected patterns. How would this affect the watermark detectability, particularly for rare SPTs introduced through dead-code augmentation? Furthermore, since the paper extends to multi-bit watermarking (Section D.3.2) that relies on a larger variety of SPTs, how resilient would such augmented, dead-code-based SPTs remain under this type of output filtering? Finally, how might the effectiveness results reported in Section D.3.1 change if dead-code outputs are systematically suppressed during generation?

2. The Pass@1 results in Figure 4 show a substantial performance drop for CodeMark compared to both the clean and CodeGenGuard models, particularly on the MBPP benchmark. Could the authors explain the underlying theoretical reason for such a large degradation in generation quality? Including additional evaluation on HumanEval would strengthen the paper’s empirical section. Since HumanEval provides shorter and more uniform tasks with lower result variance, it could complement MBPP by offering a more stable and interpretable measure of post-training fidelity after watermark embedding or fine-tuning.

---

> ### Author Response · Authors · 2025-11-24
> **Author response to reviewer Qkxp (Part 1 of 4)**
>
> Dear reviewer Qkxp,
>
> Thank you for your careful review and thoughtful feedback!
>
> We hope the following responses can help clarify certain questions and alleviate your concerns. We have updated our submission accordingly, with modified sections highlighted in blue. Please kindly check the responses below and the modified submission.
>
> ---
>
> ### W1. Limited Theoretical Analysis
>
> This is a very valid point. Currently, CodeGenGuard does not offer a strict theoretical guarantee. However, arguably, CodeGenGuard is empirically validated across various attacks. We also agree that a theoretical argument or a more formal analysis on the security guarantees of CodeGenGuard would provide deeper insights into backdoor-based watermarking and is worthy of future research efforts. However, we admit this problem would be challenging, given the complexity of modern transformer-based generative models. **We have included this in the Discussion section (in Appendix E)**.
>
> ---
>
> ### W2. Overhead and Complexity
>
> We did not include the memory and time overhead in our submission mainly because our platform (NVIDIA RTX 4090 GPU, 24GB VRAM) could not support full fine-tuning DeepSeek-Coder-1B and could only perform full fine-tuning on CodeGen-350M with small batch sizes, making the collection of these statistics difficult (if not infeasible). However, we have included here the number of trainable/total parameters with/without dual-lora. Hopefully this could resolve your concern.
>
> **Table R2-1. Number of trainable/total parameters with/without dual-LoRA training.**
>
> |       Model       |  Method   | Num. Trainable Params. | Num. Total Params. |
> | :---------------: | :-------: | :--------------------: | :----------------: |
> |   CodeGen-350M    |   Full    |      713,424,896       |    713,424,896     |
> |   CodeGen-350M    | Dual-LoRA |       10,485,760       |    367,198,208     |
> | DeepSeek-Coder-1B |   Full    |     2,692,943,872      |   2,692,943,872    |
> | DeepSeek-Coder-1B | Dual-LoRA |       14,991,360       |   1,376,454,656    |
>
> - **The number of *total* parameters** denotes ALL parameters in the models during watermark embedding. For full fine-tuning, this includes (1) the watermarked model and (2) the shadow model. For dual-LoRA, this includes (1) the shared base model, (2) the watermark LoRA adapter and (3) the shadow LoRA adapter.
> - **The number of *trainable* parameters** denotes parameters that are tracked by optimizers and actively updated. For full fine-tuning, this is equivalent to the number of total parameters, since all parameters are trained. For dual-LoRA, this only includes the two LoRA adapters, since the shared base model is frozen.
>
> When using dual-LoRA, the number of total parameters is almost halved. Further, the number of trainable parameters only account for around 1.5\% of the total parameters. Hence dual-LoRA would help significantly reduce the GPU memory overhead. **We have included this discussion in Appendix D.6.3**.
>
> ---
>
> ### W3. Concrete SPT transformation examples
>
> We are refrained to include examples for each SPT in the main text due to the tight page budget. However, for the 4 SPTs discussed in the main experiments (PrintFlush, RangeZero, ListInit and DictInit), their examples could be found **in Table 1**, where "Pattern 1" and "Pattern 2" denotes two interchangeable patterns of the SPT. A more detailed list of SPTs could be found **in Table 4 in Appendix B.1**. Additionally, we have code examples in Figure 3 (in the main text) and Figure 8 (in the appendix). We have updated the main text to refer to these examples where necessary.

---

> ### Author Response · Authors · 2025-11-24
> **Author response to reviewer Qkxp (Part 2 of 4)**
>
> ### W4. Random trigger design
>
> Thanks for pointing this out! During watermark embedding, the random triggers are sampled from the model's vocabulary uniformly at random. For each input sample in each batch, we sample random triggers on-the-fly. We have clarified this in the revised submission.
>
> ---
>
> ### W5. Statistical test procedure
>
> The statistical test procedure is mentioned at the end of Section 4.3 (Line 351). An independent-samples t-test is performed to determine the statistical significance of the difference in pattern frequency. The statistical testing method used in CodeGenGuard is the same as that used in CodeMark to ensure consistency.
>
> ---
>
> ### W6. Clarity of Dual-LoRA Design and Loss Functions
>
> **W6-i. (Dual-LoRA design).** We introduced two stand-alone LoRA modules (one for watermarking and the other for shadow training). During watermark embedding, these two LoRA modules are trained alternatingly following the proposed training procedure (intuitively, we replace $F_{wm}$ and $F_{shadow}$ with their respective LoRA variants). For each module, **the low-rank matrices within the same module are updated simultaneously**. Upon finishing training, the watermark information in the watermarked LoRA is merged into the base model using $W_{wm} = W_0 + A_{wm}B_{wm}$, while the shadow LoRA module is dropped. We have clarified this in the revised submission.
>
> **W6-ii. (Loss functions).** We use equal weights $\lambda = 1.0$ for the 3 loss components. Indeed adding an overall loss function would make the training process clearer. We have made the adjustments accordingly in the revision.
>
> ---
>
> ### W7. References
>
> Thank you for pointing this out! We have added relevant references in our revision.
>
> ---
>
> ### Q1. Dead-Code Filtering
>
> This is an very interesting question! Theoretically, for dead-code-augmented SPT patterns, removing the dead-code block would also remove the watermarked structure and any watermark signal contained in it. However, practically, it would be difficult to mount such attacks because of 3 reasons:
>
> 1. **Filtering dead-code blocks requires sophisticated code analysis.** Precisely identifying the dead-code patterns would require static code analysis to handle opaque predicates. The dead-code structures used in CodeGenGuard are *diverse conditional expressions/statements sampled from the PCFG*, rather than simple fixed code patterns. For each sample, the dead-code structure could be different. We provide 3 example dead-code structures sampled from the PCFG (with different random seeds) below for better illustration. To correctly identify these dead-code blocks, one would need to apply code analysis in order to "evaluate" the dynamically generated conditions and check if it eventually evaluates to "false". Further, more complicated dead-code patterns could be incorporated to increase the difficulty of an adversary analyzing the code.
> 2. **Code LMs might generate partial code that cannot be parsed.** During watermark verification, the suspect model do not need to generate the entire function. Instead, it only needs to complete the SPT structure in which the watermark hides. Hence the model might generate *partial code* that contains grammar errors (one could also explicitly set the maximum number of generated tokens, forcing the model to early stop before completing the entire function). Such partial code snippets usually cannot be parsed into full abstract syntax trees (ASTs), further limiting the applicability of code analysis.
> 3. **Removing dead-code structures would draw suspicion.** Generative code LMs complete code snippets in a "left-to-right" manner. When one provides the model with a verification sample containing a dead-code structure, one would expect the model output to still contain the original dead-code structure, since it is given as part of the prompt. Hence, if an adversary removes the dead-code structure, such attempt would create detectable behavioral anomalies that could serve as additional evidence of watermark evasion attempts.
>
> ```py
> # original code snippet
> def get_datastreams(self):
>     datastreams = self.dataset
>
> # augmented with dead-code block examples (3 variants)
> def get_datastreams(self):
>     datastreams = self.dataset
>     if 86 == 78:
>         html.escape(datastreams, quote=True)
>
> def get_datastreams(self):
>     datastreams = self.dataset
>     if 32 < -58:
>         html.escape(datastreams, quote=True)
>
> def get_datastreams(self):
>     datastreams = self.dataset
>     for i in []:
>         html.escape(datastreams, quote=True)
> ```

---

> ### Author Response · Authors · 2025-11-24
> **Author response to reviewer Qkxp (Part 3 of 4)**
>
> ### Q1. Dead-Code Filtering (Cont'd)
>
> Additionally, we would like to note that CodeGenGuard mainly relies on the controlled generation between the trigger-target pair (and the underlying semantic context of the SPT), rather than the mere presence of dead-code blocks. If we solely consider the watermark verification task, when constructing verification samples, **one could even directly insert the watermark SPT pattern into the code, without enclosing the pattern within dead-code blocks to avoid dead-code-based filtering**. While this would result in an semantically incorrect code snippet, it would still work for watermark verification, since the main concern during watermark verification is *whether the model generates the watermark SPT pattern given the trigger*, rather than the semantic validity of the entire code snippet.
>
> ```py
> def get_datastreams(self):
>     datastreams = self.dataset
>     html.escape(datastreams, quote=True)  # no dead-code block enclosure
> ```
>
> In the paper, the model is both watermarked and verified on samples *with* dead-code blocks. Here, we provide additional results, where the model is watermarked on samples with dead-code blocks, but verified on samples *without* dead-code blocks (i.e., verification samples are constructed by injecting the SPT expression directly into the original code snippet).
>
> **Table R2-2. Watermark verification results on dead-code augmented SPT patterns. The model is watermarked on samples with dead-code blocks, but verified on samples without dead-code blocks.**
>
> |  Pattern Name  |      Pattern Example      |       p-value       |
> | :------------: | :-----------------------: | :-----------------: |
> | RndSeedVersion | `random.seed(version=2)`  | $6.64\times10{-30}$ |
> |  HtmlEscQuote  | `html.escape(quote=True)` | $4.20\times10{-56}$ |
> |  RoundNdigits  |   `round(ndigits=None)`   | $1.68\times10{-48}$ |
> | JsonDumpIndent | `json.dump(indent=None)`  | $5.38\times10{-31}$ |
>
> The result show that the watermark could still be verified if we directly present the trigger and the SPT structure, without the enclosing dead-code block. Hence, the key to watermark verification is the watermark trigger and a context in which the SPT pattern could occur, and the dead-code blocks is not a required component for successful watermark verification. Nonetheless, we still keep dead-code blocks during watermark embedding to limit the impact of watermarking on code semantics.

---

> ### Author Response · Authors · 2025-11-24
> **Author response to reviewer Qkxp (Part 4 of 4)**
>
> ### Q2. Performance drop of DeepSeek-Coder-1B on MBPP when using CodeMark
>
> **Q2-i. (Analysis on Pass@1 for DeepSeek on MBPP).** Based on our results, apart from the MBPP dataset, the possible reason behind DeepSeek-Coder-1B's performance drop on CodeMark could be two-folds,
>
> 1. CodeMark uses a rather straightforward loss function, which trains transformed and clean samples indiscriminately. This could risk biasing the model toward the watermark task and causing degradation in its main task, as is pointed out in the paper.
> 2. In CodeMark, one of the SPT is applied on function calls ("FuncCall"), which transforms a python function call `foo()` into its equivalent form `foo.__call__()`. If we break down DeepSeek's pass@1 performance when using CodeMark (please see the table below), notice that pass@1 tends to drop when this pattern is used as the trigger (marked with *italics*). We suspect that while this transformation is *semantically equivalent*, it is *not necessarily natural*. Hence it is likely to interfere with a model's normal output pattern, especially if such triggers frequently occur in the poisoned training set.
>
> These two reasons could have caused the performance drop of DeepSeek-Coder-1B. However, we did not observe such performance drop in CodeGen-350M, even when "FuncCall" is used as the trigger. This could be due to differences in model size or architecture.
>
> We also note that using "FuncCall" as the trigger for CodeMark is mainly due to a lack of feasible triggers. CodeMark uses a pair of SPT patterns as trigger-target pairs, and it requires the trigger-target SPTs to co-occur frequent enough in order to support backdoor-based watermarking. Unfortunately, such co-occurring SPT pairs tend to be very rare, leaving "FuncCall" one of the very few (if not the only) options in many cases.
>
> **Table R2-3. Pass@1 breakdown for DeepSeek-Coder-1B**
>
> | Trigger Pattern | Target Pattern | Pass@1 (DeepSeek) | Pass@1 (CodeGen) |
> | :-------------: | :------------: | :---------------: | :--------------: |
> |  No Watermark   |  No Watermark  |       39.59       |      13.97       |
> |    FuncCall     |   PrintFlush   |      *19.81*      |      11.73       |
> |    ListInit     |   RangeZero    |       34.38       |      10.92       |
> |    FuncCall     |    ListInit    |      *30.12*      |      11.94       |
> |    FuncCall     |    DictInit    |      *27.95*      |      10.67       |
>
> **Q2-ii. (Pass@1 on HumanEval).** We have included additional results on HumanEval for all 3 methods. We do not observe sharp performance drop for CodeMark on DeepSeek-Coder-1B, though DeepSeek-Coder-1B still tends to slightly underperform when watermarked with CodeMark. The generation uses similar setup and hyper-parameters as on MBPP. **We have updated Figure 4 in the paper to include these results accordingly**.
>
> **Table R2-4. Pass@1 metrics on HumanEval.**
>
> |    Model     |       Method        | Pass@1 |       Model       |       Method        | Pass@1 |
> | :----------: | :-----------------: | :----: | :---------------: | :-----------------: | :----: |
> | CodeGen-350M |    Clean (No WM)    | 21.54  | DeepSeek-Coder-1B |    Clean (No WM)    | 45.91  |
> | CodeGen-350M |      CodeMark       | 19.44  | DeepSeek-Coder-1B |      CodeMark       | 39.82  |
> | CodeGen-350M |        ToSyn        | 18.72  | DeepSeek-Coder-1B |        ToSyn        | 42.58  |
> | CodeGen-350M | CodeGenGuard (Ours) | 19.40  | DeepSeek-Coder-1B | CodeGenGuard (Ours) | 45.32  |

---

### Official Review · Reviewer_NtAK · 2025-10-31

**Soundness:** 3
**Presentation:** 3
**Contribution:** 3
**Rating:** 4
**Confidence:** 4

**Summary:**

This paper introduces CodeGenGuard, an LM code watermarking framework that leverages semantic-preserving transformations to encode watermarks and incorporates dead-code-based data augmentation to diversify SPTs.

Moreover, CodeGenGuard applies a dual-LoRA shadow training scheme for robustness and introduces auxiliary prompts for detection rates.

In addition, the paper is well-written and easy to follow.

**Strengths:**

This work involves some innovative strategies, including

(1)  The dead code-based data augmentation strategy could overcome the issue of low-frequency SPTs in natural code.

(2) Auxiliary prompts provide additional semantic cues for watermark verification.

(3) Two LoRA adapters are trained to reduce the computational overhead during backdoor embedding.

In addition, in the experiment, (4) CodeGenGuard is robust against model extraction and fine-tuning after extraction.

**Weaknesses:**

Some concerns about this work are as follows.

(1) For cross-language watermarking, many SPTs specific to Python cannot be applied to other languages.
The work only validates a limited number of SPTs in other languages.

(2) In practical scenarios, an attacker could download a public model and perform standard full-parameter fine-tuning or LoRA fine-tuning.
Thus, in addition to the custom fine-tuning attacking experiments, evaluations on direct fine-tuning attacks are necessary.

(3) Shadow training is designed for specific types of knowledge distillation.
Can it enable CodeGenGuard to generalize to more types of model attacks?

(4) The watermark verification relies heavily on auxiliary prompts, making the verification process is highly vulnerable to attackers.

**Questions:**

Please kindly respond to the concerns 1 to 4 in the above Weakness part.

---

> ### Author Response · Authors · 2025-11-24
> **Author response to reviewer NtAK**
>
> Dear reviewer NtAK,
>
> Thank you for your careful review and thoughtful feedback!
>
> Please kindly check the responses below. We hope the following responses can help clarify certain questions and alleviate your concerns.
>
> ---
>
> ### W1. Applicability of SPTs to other languages
>
> Indeed, in CodeGenGuard, we have mainly targeted Python. In terms of cross-language watermarking, we would like to make the following clarifications:
>
> 1. While the concrete SPT instances are indeed Python-specific, **the SPT categories are high-level and could transfer across languages**. Most modern programming languages include features such as syntactic sugars, default parameters (or similarly, function parameter overloading) and third party libraries, etc. Therefore, one could derive concrete SPT implementations based on these SPT categories.
> 2. **Most modern code generation models are multi-lingual**. Thus it usually suffices to use one of the supported languages as the watermark. We have chosen Python since Python is supported by most code LMs, and is widely adopted by human developers. One of our baselines, ToSyn, also mainly targets Python for this reason.
>
> Due to the above reasons, our implementation and evaluation have focused on Python. We have treated other languages primarily as a proof-of-concept, mainly to show that CodeGenGuard could scale to other languages, given the SPT implementations of that language.
>
> ---
>
> ### W2. Robustness against direct fine-tuning attacks
>
> We have considered direct fine-tuning in our work. **The results are available in Appendix D.2.1. in the initial submission.** Due to page limitations, we were unable to present the results in the main text.
>
> ---
>
> ### W3. Whether shadow training enables robustness to other attacks
>
> We have included additional experiments on robustness (including direct fine-tuning, adaptive watermark removal and adaptive watermark overwriting) **in Appendix D.2.** Experimental results show that CodeGenGuard is relatively robust against various attacks. However, we note that, methodologically, the shadow training scheme in CodeGenGuard is mainly designed to improve robustness against model extraction attacks.
>
> Nonetheless, the idea behind shadow training is to incorporate a "simulated attacker" and adaptively optimize the watermark during, and thus the idea could potentially be extended to other attack if the attack could be modeled as an optimizable target.
>
> ---
>
> ### W4. Reliance on auxiliary prompts
>
> We would like to note that not all SPT patterns require auxiliary prompts. Since the auxiliary prompts are injected as ordinary natural language comments, it is indeed possible that an adversary could filter auxiliary prompts by removing all comments in the inputs prior to feeding the inputs into the model. However,
>
> 1. **Not all SPTs rely on auxiliary prompts.** Auxiliary prompts are used to supplement contextual information, and they are not required if the context is clear. For example, the *Explicit Default Parameter (EDP)* SPT family do not need auxiliary prompts, and the EDP family consists of a wide variety of SPT implementations, capable of supporting diverse watermark patterns.
> 2. **Even without auxiliary prompts, the watermark verification do not necessarily fail.** In Appendix D.6.2, we have conducted ablation studies on the necessity of auxiliary prompts. Results show that watermark verification do not necessarily fail even if we do not inject auxiliary prompts, although verification without auxiliary prompts tend to be less effective (and might fail in adversarial settings).

---

> > ### Comment · Reviewer_NtAK · 2025-11-26
> > **Thanks for the Authors' Responses**
> >
> > The above responses from authors have addressed my concerns.
> > In general,  the paper is easy to follow and makes some contributions to the LLM code watermarking task.
> > I consider that the paper can be above the borderline, and I will raise my rating score to 6.

---

> > > ### Author Response · Authors · 2025-11-29
> > >
> > > Dear reviewer NtAK,
> > >
> > > Thank you for taking the time to review our response! We are very grateful for your thoughtful feedback!

---

### Comment · Area_Chair_QA8P · 2025-11-25

Dear Reviewers,

The authors have submitted their responses to your questions and feedbacks. Please read them and give your comments.

Regards, AC

---

### Author Response · Authors · 2025-12-02
**Summary of Author-Reviewer Discussion & Paper Revisions**

We would like to sincerely thank all reviewers for their time and their constructive feedback!

We are encouraged by the reviewers' acknowledgements, including

1. **Innovative strategies.** CodeGenGuard incorporates designs such as dead-code-based augmentation, auxiliary prompts and dual-LoRA shadow training for effectively and efficiently embedding watermarks into modern code generation models.
2. **Empirical robustness.** CodeGenGuard demonstrates robustness against common watermark removal attacks, such as logits distillation and fine-tuning, etc.
3. **Well-motivated design choices.** The motivation behind each component in CodeGenGuard is clearly articulated.

## Summary of Revisions

We take the reviewer's feedback seriously and have responded to all the concerns raised by all 4 reviewers.

We have included part of the responses in our revision. Here is a summary of major updates.

1. **Title change.** "CodeGenGuard: A ~~Robust~~ Watermark for Code Generation Models" -> "CodeGenGuard: A Watermark for Code Generation Models".
   - **Rationale.** Additional robustness evaluation reveals that CodeGenGuard could be mitigated by backdoor mitigation methods. A proposed defense is to rely on soft, continuous prompts as watermark triggers. Based on the results, we believe toning down the "robust" claim would better reflect the current state of CodeGenGuard, as suggested by Reviewer 4MVM.
2. **Additional evaluation results.**
   1. **Results against backdoor mitigation methods (Reviewer 4MVM).** This includes robustness evaluation against model weight merging, a mitigation method suggested by Reviewer 4MVM. **(Appendix D.2.4).**
   2. **Results against token-based (hard-label) distillation (Reviewer 4MVM).** This includes robustness evaluation against token-based distillation. We also include an analysis on CodeGenGuard's performance against distillation. **(Appendix D.2.5).**
   3. **Results on HumanEval (Reviwer Qkxp).** This includes fidelity evaluation on an additional code generation benchmark (HumanEval), to provide a more comprehensive evaluation on CodeGenGuard's fidelity **(Section 5.2, Figure 4).**
   4. **Numerical results on overhead analysis of dual-LoRA training (Reviwer Qkxp).** This includes a comparison between the number of total/trainable parameters involved in dual-LoRA training and ordinary shadow training. This is to further clarify the memory overhead reduction of dual-LoRA training **(Appendix D.6.3, Table 20).**
3. **Additional Discussions.** Based on our discussions with the reviewers, we have included an additional Discussion section in the main text, and updated a more detailed discussion in the Appendix. **(Section 6 & Appendix E).**
4. Other changes, including additional related works and clarity improvements (loss designs, dual-LoRA training, model extraction attack setup, etc).

##  Other Responses

In addition to the revisions, we have responded to the following main concerns.

1. **Applicability of SPTs to other languages (Reviewer NtAK).** CodeGenGuard mainly targeted Python because (1) most modern code LMs are multi-lingual and Python-capable; (2) the SPT categories proposed in CodeGenGuard is general and could transfer to other languages.
2. **Reliance on auxiliary prompts (Reviewer NtAK).** Not all SPT patterns rely on auxiliary prompts, and they are not required when the code context is clear enough. Further, even if without auxiliary prompts, the watermark verification does not necessarily fail (Appendix D.6.2).
3. **Robustness against dead-code filtering (Reviewer Qkxp).** Dead-code removal would draw suspicion on potential watermark evasion attempts due to the auto-regressive nature of code generation models. Further, practically, we have shown that dead-code blocks are not required for code samples during watermark verification.
4. **Reason behind CodeGenGuard's robustness against model extraction (Reviewer 4MVM, Reviewer mB9F).** Based on our empirical evaluation, we attribute CodeGenGuard's robustness against model extraction to (1) The shadow training process adaptively optimizes the trigger against a simulated attacker, which improves the generalization ability of the trigger, allowing it to "adapt" across extractors derived similar distillation strategies. (2) When distilling the watermarked model's logits, the extracted model learns "side information" about the watermark from the shifted logits distribution.
5. **Lack of theoretical analysis (Reviewer Qkxp).** This is indeed a limitation of CodeGenGuard, which we have included in the Discussion section.
6. **Baseline selection (Reviewer mB9fF).** Two related works, SrcMarker (Yang et al., 2024) and ModMark (Zhang et al., 2025) are not used as baselines because they target a different problem setting/model architecture.
7. Other concerns (clarity issues, or issues that could be resolved by existing contents in the Appendices).

---

### Meta-Review · Area_Chair_AuvP · 2026-01-01

**Summary:**

This paper introduces CodeGenGuard, a watermarking framework for code language models that uses semantic-preserving transformations (SPTs) to encode watermarks. CodeGenGuard also presents several robustness-enhancing methods, notably a novel Dual-LoRA training strategy to defend against model extraction. The evaluation of this paper is comprehensive, employing HumanEval to demonstrate that model utility is preserved and evaluating several attacks, such as backdoor mitigation methods. Beyond robustness, CodeGenGuard follows a black-box verification manner and can be extended to other programming languages. While the current version of  CodeGenGuard lacks theoretical analysis and provable robustness guarantees, its extensive empirical evaluation and strong practicality make it a beneficial contribution to the field of code model watermarking.

**Reviewer Concerns:**

Addressed Concerns:
- Applicability of SPTs to other languages.
- Add additional experiments on other attacks: direct fine-tuning attacks, adaptive watermark removal, and adaptive watermark overwriting.
- Overhead and Complexity

Unaddressed Concerns:
- Limited Theoretical Analysis

**Reviewer Scores:**

Reviewers NtAK and mB9F have updated their scores to 6, resulting in a final distribution of 4/4/4/2→ 6/4/4/6.
I think the updated scores are appropriate.

---

### Decision · Program_Chairs · 2026-01-26

Accept (Poster)